# On the Regularization of Learnable Embeddings for Time Series Forecasting

**Luca Butera**                                                                     *luca.butera@usi.ch*
*Università della Svizzera Italiana, IDSIA*

**Giovanni De Felice**                                                  *g.de-felice@liverpool.ac.uk*
*University of Liverpool*

**Andrea Cini**                                                                     *andrea.cini@usi.ch*
*Università della Svizzera Italiana, IDSIA*

**Cesare Alippi**                                                                 *cesare.alippi@usi.ch*
*Università della Svizzera Italiana, IDSIA*
*Politecnico di Milano*

**Reviewed on OpenReview:** *https://openreview.net/forum?id=F5ALCh3GWG*

## Abstract

In forecasting multiple time series, accounting for the individual features of each sequence can be challenging. To address this, modern deep learning methods for time series analysis combine a shared (global) model with local layers, specific to each time series, often implemented as learnable embeddings. Ideally, these *local embeddings* should encode meaningful representations of the unique dynamics of each sequence. However, when these are learned end-to-end as parameters of a forecasting model, they may end up acting as mere sequence identifiers. Shared processing blocks may then become reliant on such identifiers, limiting their transferability to new contexts. In this paper, we address this issue by investigating methods to regularize the learning of local learnable embeddings for time series processing. Specifically, we perform the first extensive empirical study on the subject and show how such regularizations consistently improve performance in widely adopted architectures. Furthermore, we show that methods attempting to prevent the co-adaptation of local and global parameters by means of embeddings perturbation are particularly effective in this context. In this regard, we include in the comparison several perturbation-based regularization methods, going as far as periodically resetting the embeddings during training. The obtained results provide an important contribution to understanding the interplay between learnable local parameters and shared processing layers: a key challenge in modern time series processing models and a step toward developing effective foundation models for time series.

## 1 Introduction

Collections of related time series characterize many applications of learning systems in the real world, such as traffic monitoring (Li et al., 2018; Yu et al., 2018), energy analytics (Dimoulkas et al., 2019; Gasparin et al., 2022), climate modeling (Ma et al., 2023; Chen et al., 2023), and biomedical data processing (Jarrett et al., 2021; Zhang et al., 2022). The success of deep learning in the associated tasks, e.g., forecasting (Shih et al., 2019; Benidis et al., 2022), imputation (Cao et al., 2018; Cini et al., 2022) and virtual sensing (Wu et al., 2021),

relies on effectively modeling shared patterns across time series while also accounting for their individual characteristics (Benidis et al., 2022). In this context, models must rely on some attributes or positional encodings to tailor the processing to the target time series, with the risk of requiring long observation windows and high model capacity when those are not available (Salinas et al., 2020; Montero-Manso & Hyndman, 2021). Indeed, prior and positional information is often unavailable or insufficient to provide effective specialization. Notably, the study of methods to tune a shared time series model on a specific task is a major concern for the development of foundation models, i.e., backbone models trained on large collections of time series and then applied to specific target applications (Garza & Mergenthaler-Canseco, 2023; Liang et al., 2024). This problem is also particularly relevant in spatiotemporal forecasting when dealing with multiple time series coming from a sensor network (Bai et al., 2020; Cini et al., 2023b).

Among different methods, research has addressed the problem by looking into *hybrid* global-local architectures, i.e., architectures that combine global models with local learnable components (e.g., layers) specific to a target time series (Wang et al., 2019; Smyl, 2020). As a prominent example, Smyl (2020) won the M4 competition (Makridakis et al., 2020) by combining a shared *recurrent neural network* (RNN) with local exponential smoothing models fitted on each time series. In the context of spatiotemporal data, research has shifted towards the adoption of learnable embeddings (Bai et al., 2020; Shao et al., 2022), i.e., vectors of learnable parameters, to reduce the cost of learning (more complex) local processing blocks (Cini et al., 2023b). Each embedding, associated with a specific time series, is fed into the shared modeling architecture and trained end-to-end alongside it. These representations can go beyond simply encoding coordinates, as they can account for the dynamics of each time series with respect to the other sequences in the collection (Cini et al., 2023b). Parameters of this kind are analogous to the word embeddings used in natural language processing (NLP) (Mikolov et al., 2013; Peng et al., 2015). We will refer to them as *local embeddings*.

While local embeddings offer significant advantages, their adoption introduces potential drawbacks. In the first place, differently from NLP applications that operate on a fixed-size dictionary, the target time series might change over time, and new sequences might be added to the collection. Secondly, since embeddings are learned jointly with the entire architecture, co-adaptation (Srivastava et al., 2014) with the shared layers might result in embeddings being used as simple sequence identifiers (Geirhos et al., 2020). This interdependence can hinder the flexibility of the shared processing blocks and limit their transferability. Existing works (Yin et al., 2022; Cini et al., 2023b; Prabowo et al., 2024) show evidence that constraining the structure of the embedding space can lead to transferability improvements. However, no prior work has systematically addressed and evaluated regularization methods for local embeddings in forecasting architectures.

To fill this void, we investigate the impact of regularizing the learning of local embeddings for related time series within a selection of commonly used deep learning forecasting architectures. In particular, we perform an extensive empirical study comparing a variety of regularization strategies, ranging from standard approaches, such as weight penalties and dropout, to more advanced methods, e.g., clustering and variational regularization. Our analysis considers a range of scenarios, including transductive settings, transfer learning, and sensitivity analyses. Empirical results show that methods that attempt to prevent the co-adaptation of the local and global blocks by perturbing the embeddings at training time are consistently among the best-performing approaches. To further validate this observation, we include in the analysis a *forget-and-relearn* strategy (Zhou et al., 2021), named *forgetting regularization*, that periodically resets embeddings during training. The results obtained by considering this approach further show that perturbation of local parameters offers a good design principle for regularization strategies in this context. Our main findings can be summarized as follows.

(**F1**) In both transductive and transfer learning settings, regularizing the local embeddings can provide consistent performance improvements across several forecasting architectures.

(**F2**) Regularization strategies based on local parameters' perturbations aimed at preventing co-adaptation consistently lead to larger performance gains compared to other approaches.

(**F3**) Finding (**F2**) can be used as a design principle for designing new regularization strategies to mitigate overfitting and improve transferability, as highlighted by the competitive performance of the considered *forgetting* regularization.

The regularization of local learnable embeddings emerges as an often neglected but central aspect, which, with negligible computational overhead, can lead to performance improvements across all the considered benchmarks. This makes our study an important missing piece for guiding practitioners and researchers in building and designing modern neural architectures for time series processing.

## 2 Preliminaries

This section introduces the notation and formalizes the problem of time series forecasting, with a focus on the class of deep learning architectures we consider in this paper.

### 2.1 Problem settings

Consider a collection $\mathcal{D}$ of $N$ time series, where each $i$-th time series consists of a sequence of $T$ observations $\{\boldsymbol{x}_t^i \in \mathbb{R}^{d_x}\}_{t=1}^T$. Specifically, we indicate as *related time series* a set of homogenous time series coming from the same domain but generated by different sources (e.g., different sensors). Examples include sales figures for different products, or energy consumption of various users. Time series might be acquired both asynchronously and/or synchronously; in the latter case, we denote the stacked $N$ observations at time step $t$ by the matrix $\boldsymbol{X}_t \in \mathbb{R}^{N \times d_x}$. We use the shorthands $\boldsymbol{X}_{t:t+T}$ to indicate the sequence of observations within the time interval $[t, t+T)$ and $\boldsymbol{X}_{\leq t}$ to indicate those up to time $t$ (included). Eventual (exogenous) covariates associated with each time series are denoted by $\boldsymbol{u}_t^i \in \mathbb{R}^{d_u}$ ($\boldsymbol{U}_t \in \mathbb{R}^{N \times d_u}$). We focus on multistep-ahead time series forecasting, i.e., the problem of predicting the next $H$ future values for each $i$-th time series $\boldsymbol{x}_{t:t+H}^i$ given exogenous variables and a window of $W$ past observations. We focus on point forecasts, while probabilistic predictors might also be considered.

### 2.2 Hybrid global-local architectures for time series

We consider a broad class of models similar to those in (Benidis et al., 2022) and (Cini et al., 2023a). In particular, we consider predictors such as

$$\hat{\boldsymbol{x}}_{t:t+H}^i = F(\boldsymbol{x}_{t-W:t}^i, \boldsymbol{u}_{t-W:t+H}^i; \boldsymbol{\theta}), \quad i = 1, ..., N \tag{1}$$

where $\boldsymbol{\theta}$ are the learnable parameters of the model, $\hat{\boldsymbol{x}}_{t:t+H}^i$ indicates the predicted values of $\boldsymbol{x}_{t:t+H}^i$, and $F(\cdot; \boldsymbol{\theta})$ is a family of parametric forecasting models. Models in Eq. 1 do not take into account (spatial) dependencies that might exist among time series. In scenarios such as spatiotemporal forecasting, where such dependencies might be relevant to achieve accurate predictions, models that can operate on sets of (synchronous) correlated time series should be preferred. In particular, we consider a family of models

$$\widehat{\boldsymbol{X}}_{t:t+H}^{\mathcal{S}} = F(\boldsymbol{X}_{t-W:t}^{\mathcal{S}}, \mathbf{U}_{t-W:t+H}^{\mathcal{S}}; \boldsymbol{\theta}), \qquad \forall \mathcal{S} \subseteq \mathcal{D} \tag{2}$$

where $F(\cdot; \boldsymbol{\theta})$ operates on subsets $\mathcal{S}$ of the collection of time series $\mathcal{D}$, and $\boldsymbol{X}_t^{\mathcal{S}} \in \mathbb{R}^{|\mathcal{S}| \times d_x}$ is the corresponding stack of observations at time step $t$. Models of this kind can be implemented by architectures operating on sets (Zaheer et al., 2017), e.g., attention-based architectures (Vaswani et al., 2017; Grigsby et al., 2021) or *spatiotemporal graph neural networks* (STGNNs) (Jin et al., 2023; Cini et al., 2023a) based on message passing (Gilmer et al., 2017). These models can eventually account for priors on existing dependencies among time series, which could be encoded, e.g., by a graph adjacency matrix $\mathbf{A} \in \mathbb{R}^{N \times N}$ (Cini et al., 2023a). Note that models in both families (Eqs. 1 and 2) are *global*, i.e., they share parameters among all the time series being processed, with clear advantages in terms of sample efficiency (Montero-Manso & Hyndman, 2021). Most of the recent successes in applying deep learning to time series forecasting are based on the idea of implementing such global models with a neural network (Benidis et al., 2022). In the following, we will consider models belonging to the family in Eq. 2, as Eq. 1 can be seen as a special case where $|\mathcal{S}| = 1$. In particular, we consider the whole collection at once (i.e., $|\mathcal{S}| = N$) and drop the superscript $\mathcal{S}$ to simplify the notation.

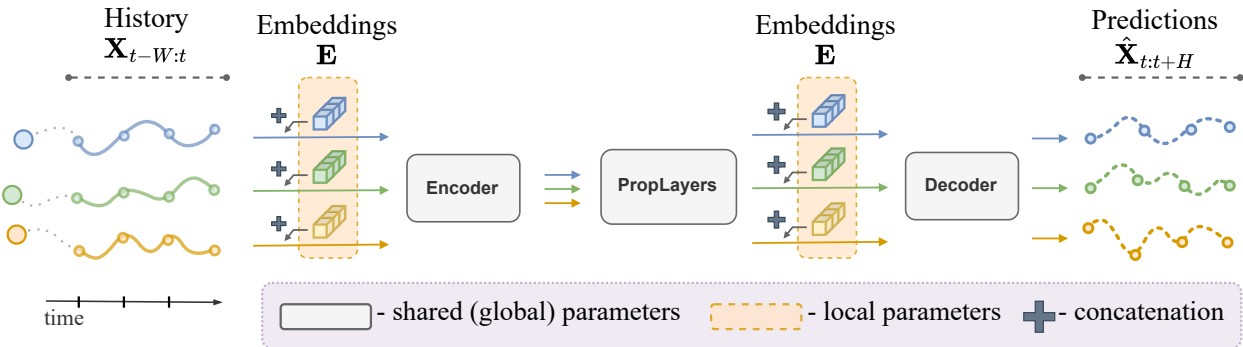

Figure 1: Overview of the hybrid global-local time series forecasting framework.

**Template architecture**  We use a template architecture analogous to that of Cini et al. (2023b). It consists of the following three processing steps: an *encoder*, one or more *propagation layers*, and a *decoder*. Predictions at each time step and time series are obtained as

$$h_t^{i,0} = \text{ENCODER}\left(\boldsymbol{x}_{t-1}^i, \boldsymbol{u}_{t-1}^i\right) \tag{3}$$

$$\boldsymbol{H}_t^{l+1} = \text{PROPLAYER}^l\left(\boldsymbol{H}_{\leq t}^l\right), \quad l = 0, 1, ..., L-1 \tag{4}$$

$$\hat{\mathbf{x}}_{t:t+H}^i = \text{DECODER}\left(\boldsymbol{h}_t^{i,L}, \boldsymbol{u}_{t:t+H}\right) \tag{5}$$

where the additional index $l$ refers to the layer depth, $\boldsymbol{h}_t^{i,l} \in \mathbb{R}^{d_h}$ is the representation associated with the $i$-th time series at the $l$-th layer, and $\boldsymbol{H}_t^l \in \mathbb{R}^{|\mathcal{S}| \times d_h}$ the stack of such representations. The ENCODER $(\cdot)$ and DECODER $(\cdot)$ blocks can be implemented in different ways, e.g. a linear layer or a multilayer perceptron (MLP), and operate on single time steps and time series. Conversely, the PROPLAYER $(\cdot)$ blocks are the only components of the architecture propagating representations across the temporal dimension and across time series in the collection. Each PROPLAYER $(\cdot)$ can be implemented by any existing sequence modeling architecture, e.g., RNNs (Hochreiter & Schmidhuber, 1997; Cho et al., 2014a) or temporal convolutional networks (TCNs) (LeCun & Bengio, 1998; Bai et al., 2018), and/or spatiotemporal operators, e.g., *spatiotemporal attention* (STAtt) models (Grigsby et al., 2021; Deihim et al., 2023). In particular, STGNNs (Seo et al., 2018; Yu et al., 2018), i.e., models that combine sequence modeling architectures with message passing operators, are among the most popular architectures for implementing propagation layers when processing correlated time series (Jin et al., 2023).

**Hybrid global-local architectures and local embeddings**  As anticipated in the introduction, despite their advantages, global models might struggle to account for the specific dynamics of each time series. Hybrid global-local architectures (Benidis et al., 2022) address this issue by including parameters specific to each target time series. If we indicate these parameters as $\Phi = \{\boldsymbol{\phi}^1, ..., \boldsymbol{\phi}^N\}$, the resulting model family would provide forecasts for the time series in the collection as

$$\widehat{\boldsymbol{X}}_{t:t+H}^{\mathcal{S}} = F\left(\boldsymbol{X}_{t-W:t}^{\mathcal{S}}, \boldsymbol{U}_{t-W:t+H}^{\mathcal{S}}; \boldsymbol{\theta}, \Phi^{\mathcal{S}}\right). \tag{6}$$

Although many implementations of such models exist (Benidis et al., 2022; Cini et al., 2023b), we consider models where the local components are implemented as embeddings $\boldsymbol{E} \in \mathbb{R}^{N \times d_e}$ of learnable parameters such that $\Phi = \boldsymbol{E}$ and $\boldsymbol{\phi}^i = \boldsymbol{e}^i$. In particular, each local embedding $\mathbf{e}^i$ is associated with the corresponding $i$-th time series and can be learned end-to-end jointly with the shared network weights. Embeddings are incorporated into the template architecture at both the encoder and decoder level by concatenating them to the input as

$$\boldsymbol{h}_t^{i,0} = \text{ENCODER}\left(\boldsymbol{x}_{t-1}^i \| \mathbf{e}^i, \boldsymbol{u}_{t-1}^i\right), \quad \hat{\mathbf{x}}_{t:t+H}^i = \text{DECODER}\left(\boldsymbol{h}_t^{i,L} \| \mathbf{e}^i, \boldsymbol{u}_{t:t+H}\right). \tag{7}$$

Fig. 1 provides an overview of the resulting reference architecture. The addition of these parameters comes at a cost in terms of flexibility, as processing time series that were not observed at training time requires fitting new parameters (Januschowski et al., 2020; Cini et al., 2023b).

## 3    Related works

Learnable embeddings are key components of state-of-the-art time series processing architectures such as STGNNs (Bai et al., 2020; Cini et al., 2023a) and attention-based models (Marisca et al., 2022; Liu et al., 2023; Xiao et al., 2024). In particular, Cini et al. (2023b) systematically addresses the role of such embeddings in hybrid global-local STGNNs. Aside from modeling local dynamics, embeddings have been used extensively to amortize the cost of learning the full adjacency matrix in graph-based models (Wu et al., 2019; Shang et al., 2021; Satorras et al., 2022; De Felice et al., 2024), or as spatial positional encodings (Marisca et al., 2022; Shao et al., 2022; Liu et al., 2023). They are also routinely used in NLP as word embeddings (Mikolov et al., 2013) and often as spatial positional encodings (Devlin et al., 2019; Yang et al., 2019).

Methods to regularize learning architectures are obviously central to deep and machine learning in general (Ying, 2019; Tian & Zhang, 2022). Traditional examples include L1 (lasso) (Tibshirani, 1996) and L2 (ridge, weight decay) (Krogh & Hertz, 1991) regularizations. Established approaches in deep learning encompass *dropout* (Srivastava et al., 2014), *layer normalization* (Ba et al., 2016) and *weight normalization* (Salimans & Kingma, 2016). Notably, several regularizations have been tailored to specific architectures (Zaremba et al., 2014; Gal & Ghahramani, 2016; Dieng et al., 2018; Wang & Niepert, 2019; Santos & Papa, 2022), and designed to improve transferability (Wang et al., 2018; Takada & Fujisawa, 2020; Abuduweili et al., 2021). Similarly in spirit to our work, Peng et al. (2015) presents a comparative study of different regularizations for word embeddings in NLP. Regarding regularizations for learnable embeddings for time series, Yin et al. (2022) uses learned embedding clusters to regularize fine-tuning on target data, facilitating transfer in graph-based forecasting. Similarly, Cini et al. (2023b) uses a cluster-based regularization, as well as a variational-based approach. However, these works only address model transferability, without considering the impact of local embedding regularization in a broader context, e.g., transductive settings.

## 4    Regularization strategies for local embeddings

In this section, we discuss some possible shortcomings of hybrid global-local forecasting architectures and present the regularization strategies employed in our experimental analysis.

Global models are inherently less likely to overspecialize to individual sequences since all parameters are shared among time series. The introduction of local embeddings reduces the regularization effect brought by this inductive bias and jointly learning local and global parameters end-to-end on a downstream task may result in a model that is more likely to overfit to individual time series, with a potential negative impact on both performance and transferability (Cini et al., 2023b). In the following, we use the term *co-adaptation* to refer to these overfitting phenomena coming from training the local embeddings end-to-end with the global model.

Model regularization is a common approach to deal with overfitting and to control model and sample complexity. For instance, one might apply well-known techniques, such as *weight decay* (Krogh & Hertz, 1991) or *dropout* (Srivastava et al., 2014), across the entire network. However, in many scenarios, the global processing blocks are not particularly overparametrized, as weights are shared, and regularization might not be necessary. In global-local models, one could consider regularizing the local parameters only, i.e., in our settings, constraining how embeddings are learned. Fig. 2 empirically supports this intuition with an illustrative example: given a global time-series forecasting model with fixed hyperparameters (*green*), adding local embeddings (*orange*) improves performance, but makes it more likely to overfit (as model complexity increases). Regularizing the entire global-local model (*purple*), however, hinders performance

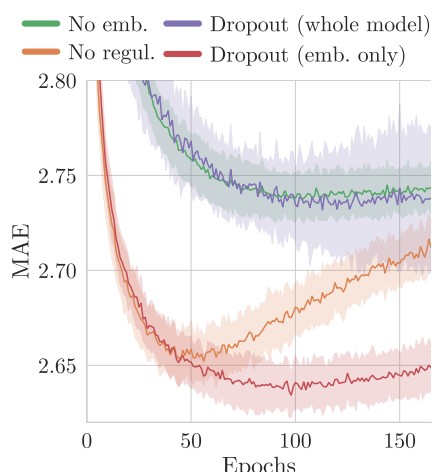

Figure 2: Validation curves when regularizing whole model or just local parameters in a time series forecasting task (STGNN model, METR-LA dataset, 50 runs, $\pm 1 std$). Models and datasets are discussed in Sec. 5.

in this scenario. Conversely, applying regularization (dropout in this case) to the embeddings only (*red*), regularizes the model effectively. Note that regularization of the global parameters can obviously still be beneficial in the case of an over-parametrized global model.

## 4.1 Regularization methods

In the following, we present the regularization methods that will later be part of our experimental analysis in Sec. 5. They range from adaptations of standard approaches to recent contributions from the literature. We also consider an approach based on parameter resetting.

**L1 and L2 regularization**   L2 regularization (Krogh & Hertz, 1991), also known as *weight decay*, consists in adding to the loss a penalty term proportional to the magnitude of the model's weights. When applied to the embeddings, the penalty term is $\mathcal{L}_{l2}\left(\boldsymbol{E}\right) = \lambda_{l2} \cdot \|\boldsymbol{E}\|_2^2$. Similarly, L1 regularization (Tibshirani, 1996), also known as *lasso*, consists in applying a penalty term of $\mathcal{L}_{l1}\left(\boldsymbol{E}\right) = \lambda_{l1} \cdot \|\boldsymbol{E}\|_1$. The positive scalar values, $\lambda_{l2}$ and $\lambda_{l1}$, control the regularization strength.

**Dropout**   *Dropout* (Srivastava et al., 2014) is another widely recognized regularization. This consists in randomly masking out individual neurons during training, with probability $p$, while scaling the others by $\frac{1}{1-p}$. This technique heuristically forces the network to learn robust representations, often preventing overfitting. In our context, we apply it to the embedding vectors by randomly masking out some parameters for each embedding.

**Clustering**   The clustering regularization introduced in (Cini et al., 2023b) is based on learning a set of cluster centroids and a cluster assignment matrix, alongside the embeddings. A regularization term, with weight $\lambda_{clst}$, is then added to the loss to minimize the distance between embeddings and the assigned centroid. It was originally designed to structure the embedding space and improve model transferability.

**Variational regularization**   *Variational regularization* (Cini et al., 2023b), consists in modeling embeddings as samples from a Gaussian posterior learned from data $\mathbf{e}^i \sim \mathcal{N}\left(\boldsymbol{\mu}_{var}^i, diag\left(\boldsymbol{\sigma}_{var}^i\right)\right)$, where $\boldsymbol{\mu}_{var}^i$ and $\boldsymbol{\sigma}_{var}^i$ are the learnable local parameters. A penalty term based on the KL-divergence between the learned distribution and a standard Gaussian prior, with weight $\lambda_{var}$, is added to the loss. Similarly to the *clustering* regularization, it was introduced to enhance model transferability.

**Forgetting**   We also consider a novel *forgetting regularization*, based on the *forget-and-relearn* paradigm (Zhou et al., 2021). This refers to strategies where the parameters of some neural network layers are occasionally reset during training. By doing this, it is possible to reduce memorization of training samples (Baldock et al., 2021), avoiding shortcuts and learning representations that generalize better (Geirhos et al., 2020; Zhou et al., 2021). We adopt this paradigm in the context of hybrid global-local time series forecasting architectures and propose the following procedure. Every $K$ training epochs, we periodically reset local embeddings $\boldsymbol{E}$ to a sample from a shared initialization distribution $\mathcal{P}_e$. Concurrently, we similarly reset the ENCODER's and DECODER's weights (Eq. 7) that directly multiply the embeddings (see Appendix C.1). The value of $K$ can be easily selected empirically (see Appendix C.2). Additionally, resetting is halted after a certain amount of epochs to allow for convergence to a final configuration; this can be easily tuned manually, by considering the model's convergence speed or triggered automatically, by monitoring the validation loss. Note that resetting local parameters to regularize the training of a related time series forecasting model has never been explored in the literature.

## 4.2 Preventing co-adaptation

As mentioned in Sec. 4, learning local and global parameters jointly on a downstream task can result in overfitting. We use the term *co-adaptation* to indicate this phenomenon, by referring to global and local parameters *co-adapting* during training. Specifically, local embeddings might simply act as sequence identifiers and result in models that rely entirely on this identification mechanism. Besides the negative effects on sample efficiency, the resulting architecture would likely lose flexibility in terms of the transferability of the

learned representations and processing blocks. Furthermore, this behavior is in contrast with the intended one for local embeddings, i.e., encoding meaningful characteristics of the target time series. One way in which regularization methods can aim to prevent such co-adaptation is to actively perturb the local parameters during training so that subsequent layers cannot rely on specific feature values (Srivastava et al., 2014). In our case, the global processing block would be less likely to rely on specific values of the embeddings when processing the target time series. In the following, we will use the term perturbation in a broad sense, referring to strategies that modify the values of local parameters at training time, e.g., resampling, zeroing, or adding noise.

Among the considered regularization methods, dropout perturbs the embeddings by randomly zeroing out parameters, as similarly done for the inputs/outputs of subsequent layers to avoid the co-adaptation of the associated weights (Srivastava et al., 2014). The variational regularization, given a common prior, learns each embedding's sampling distribution end-to-end while resampling their actual values before providing them as inputs to the downstream at each forward pass. Finally, forgetting regularization introduces perturbation in the early stages of training by periodically resetting the local parameters. In particular, by considering this strategy for the first time in this context, we aim to test whether actively perturbing local parameters offers an effective principle for the design of new regularizations for hybrid time series forecasting models.

Regarding the other regularization methods considered in the analysis, L1 and L2 regularizations simply penalize the embeddings' magnitude, while clustering similarly penalizes the distance of the embeddings from the learned centroids, forcing them to occupy specific regions within the embedding space. While these regularizations can provide structure to the embedding space, they do not actively perturb embeddings' values. The empirical results presented in Sec. 5 altogether suggest that strategies based on embedding perturbations are more effective at preventing the kind of overfitting we previously discussed, i.e., co-adaptation.

## 5 Experiments

We evaluate the effectiveness of different regularization strategies for local embeddings under three different scenarios: time series forecasting benchmarks (Sec. 5.1), transfer learning (Sec. 5.3), and a sensitivity analysis through embedding perturbations (Sec. 5.4). We consider six real-world datasets of time series collections, spanning four different application domains: **METR-LA** and **PEMS-BAY** (Li et al., 2018) as two established benchmarks for traffic forecasting, **AQI** (Zheng et al., 2015) from the air quality domain, **CER-E** (CER, 2016) from the energy consumption domain, **CLM-D** (De Felice et al., 2024) and **EngRAD** (Marisca et al., 2024) as two multivariate climatic datasets. Details on the datasets, data splits and forecasting settings can be found in Appendix A.

Regarding the investigated models, we consider three different hybrid global-local architectures (Sec. 2.2) distinguished by three different implementations of the propagation layer (Eq. 4). In particular: **1)** a RNN with gated recurrent units (GRUs) cells (Cho et al., 2014b) (**RNN**), **2)** a STGNN stacking GRU and anisotropic message-passing layers (Bresson & Laurent, 2017) (**STGNN**), and **3)** a GRU followed by multi-head attention across sequences (Vaswani et al., 2017) (**STAtt**). Such hybrid architectures are representative of the current state of the art in the considered benchmarks and of the dominant deep learning frameworks for processing sets of related time series (Benidis et al., 2022; Cini et al., 2023a). Implementation and architectural details are provided in Appendix B, while details on the experimental settings can be found in Appendix D.

### 5.1 Time series forecasting benchmarks

In our first experiment, we consider the problem of time series forecasting in a transductive setting, i.e., the set of time series to be forecast is the same set observed at training time. The forecasting horizon is dataset-dependent and reported in Tab. 4 of Appendix A. We use models without local parameters and with un-regularized local embeddings as reference baselines. Then, we evaluate performance with regularized local embeddings, adopting the different strategies detailed in Sec. 4.1. For model selection, a hyperparameter search on the hidden size (i.e., the number of units in each model's hidden layer) and learning rate has been carried out independently for each model variant and dataset. All regularization hyperparameters have been set as detailed in Appendix D.

Table 1: Forecasting test error under optimal model size and learning rate (5 runs, $\pm 1 std$). Methods equal to or better than the corresponding standard architectures (RNN, STGNN, STAtt) with embeddings (+ EMB.) are in bold. The best-performing method within each dataset and model is in red. '+ REGULARIZATION' denotes the addition of that specific regularization only, on top of '+ EMB.' .

| DATASET | METR-LA | PEMS-BAY | CER-E | AQI | CLM-D | ENGRAD |
|---|---|---|---|---|---|---|
| MODEL | MAE ↓ | MAE ↓ | MAE ↓ | MAE ↓ | MMRE ↓ | MMRE ↓ |
| RNN | $3.556_{\pm.004}$ | $1.774_{\pm.002}$ | $0.4453_{\pm.0010}$ | $13.279_{\pm.042}$ | $19.66_{\pm.01}$ | $31.04_{\pm.04}$ |
| + EMB. | $3.148_{\pm.011}$ | $1.593_{\pm.004}$ | $0.4146_{\pm.0025}$ | $13.247_{\pm.044}$ | $19.41_{\pm.01}$ | $31.01_{\pm.10}$ |
| + L1 | $3.149_{\pm.007}$ | $\mathbf{1.590}_{\pm.002}$ | $\mathbf{0.4079}_{\pm.0023}$ | $\mathbf{13.147}_{\pm.047}$ | $19.47_{\pm.01}$ | $31.02_{\pm.11}$ |
| + L2 | $\mathbf{3.146}_{\pm.009}$ | $\mathbf{1.586}_{\pm.005}$ | $\mathbf{0.4058}_{\pm.0011}$ | $\mathbf{13.181}_{\pm.026}$ | $19.42_{\pm.01}$ | $\mathbf{30.97}_{\pm.06}$ |
| + CLUST. | $\mathbf{3.138}_{\pm.015}$ | $\mathbf{1.588}_{\pm.007}$ | $0.4115_{\pm.0025}$ | $\mathbf{13.231}_{\pm.044}$ | $\mathbf{19.41}_{\pm.01}$ | $31.05_{\pm.07}$ |
| + DROP. | $\mathbf{3.147}_{\pm.012}$ | $1.580_{\pm.007}$ | $0.4104_{\pm.0005}$ | $\mathbf{13.114}_{\pm.038}$ | $19.43_{\pm.01}$ | $\mathbf{30.99}_{\pm.09}$ |
| + VARI. | $3.132_{\pm.006}$ | $\mathbf{1.589}_{\pm.006}$ | $\mathbf{0.4050}_{\pm.0006}$ | $13.113_{\pm.029}$ | $19.39_{\pm.00}$ | $\mathbf{30.98}_{\pm.05}$ |
| + FORG. | $3.149_{\pm.007}$ | $\mathbf{1.590}_{\pm.007}$ | $0.4049_{\pm.0007}$ | $\mathbf{13.185}_{\pm.022}$ | $19.42_{\pm.02}$ | $30.92_{\pm.02}$ |
| STGNN | $3.239_{\pm.017}$ | $1.660_{\pm.003}$ | $0.4275_{\pm.0006}$ | $11.814_{\pm.051}$ | $19.19_{\pm.03}$ | $28.04_{\pm.08}$ |
| + EMB. | $3.027_{\pm.009}$ | $1.593_{\pm.004}$ | $0.4144_{\pm.0032}$ | $11.881_{\pm.053}$ | $18.89_{\pm.04}$ | $27.52_{\pm.09}$ |
| + L1 | $3.040_{\pm.016}$ | $\mathbf{1.587}_{\pm.005}$ | $\mathbf{0.4039}_{\pm.0009}$ | $\mathbf{11.789}_{\pm.043}$ | $18.92_{\pm.03}$ | $\mathbf{27.45}_{\pm.12}$ |
| + L2 | $\mathbf{3.023}_{\pm.009}$ | $\mathbf{1.582}_{\pm.003}$ | $\mathbf{0.4016}_{\pm.0014}$ | $\mathbf{11.795}_{\pm.025}$ | $\mathbf{18.87}_{\pm.04}$ | $\mathbf{27.44}_{\pm.14}$ |
| + CLUST. | $\mathbf{3.025}_{\pm.012}$ | $\mathbf{1.580}_{\pm.005}$ | $0.4075_{\pm.0020}$ | $\mathbf{11.876}_{\pm.053}$ | $18.85_{\pm.04}$ | $27.60_{\pm.14}$ |
| + DROP. | $3.036_{\pm.011}$ | $\mathbf{1.575}_{\pm.006}$ | $\mathbf{0.4042}_{\pm.0008}$ | $11.712_{\pm.016}$ | $18.93_{\pm.04}$ | $27.41_{\pm.06}$ |
| + VARI. | $3.013_{\pm.005}$ | $1.566_{\pm.003}$ | $0.3989_{\pm.0012}$ | $\mathbf{11.768}_{\pm.026}$ | $\mathbf{18.85}_{\pm.02}$ | $27.53_{\pm.10}$ |
| + FORG. | $3.050_{\pm.017}$ | $\mathbf{1.578}_{\pm.006}$ | $\mathbf{0.4026}_{\pm.0006}$ | $\mathbf{11.793}_{\pm.040}$ | $18.83_{\pm.02}$ | $\mathbf{27.47}_{\pm.12}$ |
| STATT | $3.538_{\pm.004}$ | $1.776_{\pm.002}$ | $0.4479_{\pm.0014}$ | $13.341_{\pm.330}$ | $19.74_{\pm.02}$ | $29.25_{\pm.09}$ |
| + EMB. | $3.074_{\pm.020}$ | $1.616_{\pm.012}$ | $0.4143_{\pm.0022}$ | $12.973_{\pm.402}$ | $19.10_{\pm.03}$ | $28.32_{\pm.08}$ |
| + L1 | $\mathbf{3.061}_{\pm.021}$ | $\mathbf{1.590}_{\pm.005}$ | $\mathbf{0.4072}_{\pm.0009}$ | $\mathbf{12.799}_{\pm.697}$ | $19.27_{\pm.04}$ | $\mathbf{28.15}_{\pm.33}$ |
| + L2 | $\mathbf{3.058}_{\pm.017}$ | $\mathbf{1.593}_{\pm.004}$ | $\mathbf{0.4061}_{\pm.0004}$ | $13.542_{\pm.606}$ | $19.17_{\pm.06}$ | $\mathbf{28.12}_{\pm.19}$ |
| + CLUST. | $\mathbf{3.061}_{\pm.021}$ | $\mathbf{1.590}_{\pm.008}$ | $0.4084_{\pm.0017}$ | $\mathbf{12.535}_{\pm.240}$ | $19.06_{\pm.01}$ | $\mathbf{27.96}_{\pm.17}$ |
| + DROP. | $\mathbf{3.058}_{\pm.005}$ | $1.565_{\pm.004}$ | $\mathbf{0.4068}_{\pm.0012}$ | $12.443_{\pm.482}$ | $19.30_{\pm.08}$ | $\mathbf{28.12}_{\pm.23}$ |
| + VARI. | $3.041_{\pm.009}$ | $\mathbf{1.571}_{\pm.005}$ | $0.4030_{\pm.0014}$ | $\mathbf{12.616}_{\pm1.145}$ | $\mathbf{19.10}_{\pm.06}$ | $\mathbf{28.01}_{\pm.13}$ |
| + FORG. | $\mathbf{3.058}_{\pm.011}$ | $\mathbf{1.579}_{\pm.003}$ | $\mathbf{0.4062}_{\pm.0012}$ | $\mathbf{12.455}_{\pm.416}$ | $19.16_{\pm.02}$ | $27.94_{\pm.14}$ |

Tab. 1 shows the obtained results, while Tab. 2 provides a concise summary of the performance of each regularization across experimental settings. Regardless of the strategy, regularizing the learning of local embeddings provides consistent performance improvements over non-regularized models in most datasets (**F1**). Considering the negligible computational overhead and ease of implementation, these results support the adoption of such techniques as standard practice. While there is no clear winner among the different regularization methods, the *variational* regularization appears to be the most effective, on average; followed by *forgetting* and *dropout*. This suggests that methods perturbing the embedding values can be more effective in preventing co-adaptation (**F2**) (**F3**). We can speculate that the effectiveness of the *variational* regularization stems from it jointly perturbing embeddings' values while also providing structure to the embedding space. Additional results on the impact of regularizing embeddings when Eq. 4 is implemented by a simple MLP are reported in Appendix E, while results regarding the combination of multiple regularization techniques can be found in Appendix F.

Table 2: Summary statistics for Tab. 1. Columns indicate the average improvement (%) over the reference unregularized model and mean relative rank. Best is in red, second in bold.

| REG. | %IMPR. | RANK |
|---|---|---|
| L1 | 0.58 | 4.6 |
| L2 | 0.45 | 3.4 |
| CLUST. | 0.67 | 4.0 |
| DROP. | 0.94 | 3.2 |
| VARI. | 1.21 | 2.0 |
| FORG. | **0.95** | **3.0** |

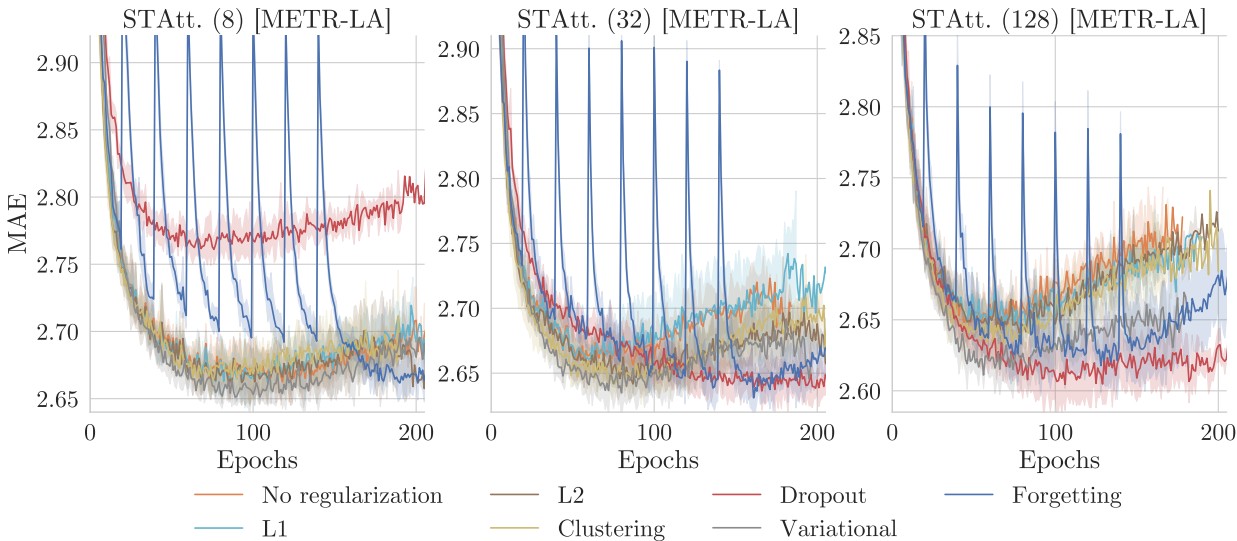

Figure 3: Validation curves for different training scenarios (5 runs, $\pm 1 std$). Plot names follow the convention *model (embedding size) [dataset]*.

## 5.2 Sensitivity analysis and learning curves

To provide additional insight, we investigate how different regularizations affect the learning curve across different embedding sizes. In doing so, we fix the shared model hidden size to $d_h = 64$ and learning rate to $lr = 0.00075$. Specifically, Fig. 3 shows the validation mean absolute error (MAE) across training epochs for some example scenarios; different colors correspond to different regularization strategies. For completeness, additional plots in complementary settings are reported in Appendix G. Comparing the different regularizations, we can see that dropout (red) and forgetting (blue) significantly affect the learning curves, even with large embeddings (Fig. 3, right). However, dropout, as one could expect, might be problematic when applied to embeddings of limited size (Fig. 3, left). The forgetting regularization is robust across the different configurations. The variational regularization appears less disruptive of the learning dynamics. The remaining regularizations (i.e., L1, L2, and clustering) have little impact on the learning curve. Overall, regularization strategies that perturb the local embeddings seem more likely to positively affect the learning curves, which might be a reason for the superior performance shown in Tab. 2.

## 5.3 Transfer learning

Our second main experiment consists of time series forecasting under a transfer learning setting, adapted from previous works (Cini et al., 2023b). This benchmark aims to verify the impact of regularizing the local embeddings on the transferability of the global (shared) processing blocks. We consider the 4 PEMS benchmark datasets from Guo et al. (2021) (i.e., PEMS03, PEMS04, PEMS07, PEMS08) and a reference STGNN model with local embeddings. For each subset of three datasets, we first train the entire model, then we reset the embeddings to their initial values and fine-tune them exclusively on the held-out dataset. During fine-tuning, shared parameters are kept frozen. The different regularizations are applied only during the initial training procedure and disabled during fine-tuning. Tab. 3 reports the results for the un-regularized and regularized models as the amount of data for fine-tuning varies (zero-shot refers to no fine-tuning at all). It is evident how dropout, variational regularization, and forgetting are the most reliable and effective strategies (**F2**). Notably, dropout excels in the zero-shot setting, while forgetting is consistent across fine-tuning lengths and excels when more data are available (**F3**). This suggests dropout can be particularly useful when data from the transfer domain is scarce or absent. Conversely, regularizations based on structuring the embedding space, i.e., L1, L2, and the *clustering* regularization, show mixed results compared to the unconstrained model.

Table 3: Forecasting test error (MAE) in transfer learning (5 runs, $\pm 1std$, STGNN). Within each target dataset, different rows pertain to different fine-tuning data lengths. Bold denotes methods equal to or better than the baseline (+ EMB.), best method is in red. '+ REGULARIZATION' denotes the addition of that specific regularization only, on top of '+ EMB.' .

| | TR. SIZE | + EMB. | + L1 | + L2 | + CLUST. | + DROPOUT | + VARIA. | + FORGET. |
|---|---|---|---|---|---|---|---|---|
| **PEMS03** | ZERO-SHOT | $26.19_{\pm.98}$ | $\mathbf{26.06_{\pm.61}}$ | $\mathbf{25.26_{\pm.68}}$ | $\mathbf{25.18_{\pm.40}}$ | $\mathbf{\color{red}21.45_{\pm.26}}$ | $\mathbf{22.71_{\pm.13}}$ | $\mathbf{25.08_{\pm.46}}$ |
| | 1 DAY | $18.92_{\pm.08}$ | $\mathbf{18.89_{\pm.10}}$ | $\mathbf{18.87_{\pm.08}}$ | $\mathbf{18.77_{\pm.05}}$ | $\mathbf{\color{red}17.99_{\pm.02}}$ | $\mathbf{18.17_{\pm.07}}$ | $\mathbf{18.55_{\pm.09}}$ |
| | 3 DAYS | $18.41_{\pm.04}$ | $\mathbf{18.41_{\pm.07}}$ | $\mathbf{18.35_{\pm.05}}$ | $18.53_{\pm.06}$ | $\mathbf{\color{red}17.86_{\pm.05}}$ | $\mathbf{17.91_{\pm.03}}$ | $\mathbf{18.19_{\pm.03}}$ |
| | 1 WEEK | $17.53_{\pm.02}$ | $\mathbf{17.52_{\pm.05}}$ | $\mathbf{17.47_{\pm.07}}$ | $17.59_{\pm.06}$ | $\mathbf{17.31_{\pm.04}}$ | $\mathbf{\color{red}17.26_{\pm.02}}$ | $\mathbf{17.34_{\pm.03}}$ |
| | 2 WEEKS | $17.34_{\pm.03}$ | $\mathbf{17.34_{\pm.03}}$ | $\mathbf{17.28_{\pm.04}}$ | $17.37_{\pm.04}$ | $\mathbf{17.20_{\pm.02}}$ | $\mathbf{\color{red}17.16_{\pm.02}}$ | $\mathbf{17.20_{\pm.05}}$ |
| **PEMS04** | ZERO-SHOT | $29.23_{\pm.43}$ | $\mathbf{29.09_{\pm.26}}$ | $29.26_{\pm.47}$ | $30.03_{\pm.26}$ | $\mathbf{\color{red}26.13_{\pm.31}}$ | $\mathbf{27.56_{\pm.31}}$ | $\mathbf{28.44_{\pm.34}}$ |
| | 1 DAY | $23.93_{\pm.16}$ | $24.00_{\pm.10}$ | $23.96_{\pm.16}$ | $\mathbf{23.71_{\pm.05}}$ | $\mathbf{\color{red}23.04_{\pm.04}}$ | $\mathbf{23.33_{\pm.12}}$ | $\mathbf{23.54_{\pm.13}}$ |
| | 3 DAYS | $23.18_{\pm.12}$ | $23.22_{\pm.14}$ | $23.27_{\pm.10}$ | $\mathbf{22.97_{\pm.06}}$ | $\mathbf{\color{red}22.67_{\pm.03}}$ | $\mathbf{22.87_{\pm.06}}$ | $\mathbf{22.86_{\pm.08}}$ |
| | 1 WEEK | $22.47_{\pm.07}$ | $\mathbf{22.46_{\pm.05}}$ | $\mathbf{22.45_{\pm.07}}$ | $22.49_{\pm.07}$ | $\mathbf{22.29_{\pm.03}}$ | $\mathbf{22.38_{\pm.05}}$ | $\mathbf{\color{red}22.26_{\pm.03}}$ |
| | 2 WEEKS | $21.92_{\pm.04}$ | $21.93_{\pm.04}$ | $21.93_{\pm.05}$ | $22.00_{\pm.01}$ | $21.96_{\pm.03}$ | $22.04_{\pm.04}$ | $\mathbf{\color{red}21.79_{\pm.03}}$ |
| **PEMS07** | ZERO-SHOT | $57.40_{\pm6.70}$ | $\mathbf{55.30_{\pm3.97}}$ | $\mathbf{53.97_{\pm3.37}}$ | $\mathbf{56.92_{\pm1.62}}$ | $\mathbf{\color{red}34.19_{\pm.78}}$ | $\mathbf{42.99_{\pm2.83}}$ | $\mathbf{42.67_{\pm2.64}}$ |
| | 1 DAY | $29.61_{\pm.17}$ | $29.63_{\pm.24}$ | $\mathbf{29.37_{\pm.27}}$ | $\mathbf{29.00_{\pm.32}}$ | $\mathbf{\color{red}27.25_{\pm.12}}$ | $\mathbf{28.30_{\pm.41}}$ | $\mathbf{28.38_{\pm.19}}$ |
| | 3 DAYS | $27.65_{\pm.14}$ | $27.74_{\pm.14}$ | $\mathbf{27.56_{\pm.14}}$ | $\mathbf{27.56_{\pm.14}}$ | $\mathbf{\color{red}26.37_{\pm.10}}$ | $\mathbf{26.92_{\pm.21}}$ | $\mathbf{26.92_{\pm.08}}$ |
| | 1 WEEK | $26.60_{\pm.04}$ | $26.66_{\pm.09}$ | $\mathbf{26.60_{\pm.07}}$ | $26.90_{\pm.14}$ | $\mathbf{\color{red}25.87_{\pm.13}}$ | $\mathbf{26.20_{\pm.17}}$ | $\mathbf{26.03_{\pm.06}}$ |
| | 2 WEEKS | $25.84_{\pm.04}$ | $25.90_{\pm.07}$ | $\mathbf{25.76_{\pm.04}}$ | $26.31_{\pm.24}$ | $\mathbf{25.53_{\pm.15}}$ | $\mathbf{25.66_{\pm.17}}$ | $\mathbf{\color{red}25.41_{\pm.04}}$ |
| **PEMS08** | ZERO-SHOT | $25.41_{\pm.62}$ | $\mathbf{25.26_{\pm1.09}}$ | $25.90_{\pm.47}$ | $25.60_{\pm.44}$ | $\mathbf{\color{red}21.04_{\pm.17}}$ | $\mathbf{22.60_{\pm.37}}$ | $\mathbf{23.90_{\pm.43}}$ |
| | 1 DAY | $18.91_{\pm.09}$ | $\mathbf{18.88_{\pm.05}}$ | $\mathbf{18.85_{\pm.08}}$ | $\mathbf{18.64_{\pm.08}}$ | $\mathbf{\color{red}17.96_{\pm.06}}$ | $\mathbf{18.23_{\pm.12}}$ | $\mathbf{18.40_{\pm.13}}$ |
| | 3 DAYS | $18.11_{\pm.10}$ | $18.16_{\pm.13}$ | $18.15_{\pm.08}$ | $\mathbf{18.03_{\pm.04}}$ | $\mathbf{\color{red}17.55_{\pm.06}}$ | $\mathbf{17.77_{\pm.07}}$ | $\mathbf{17.83_{\pm.14}}$ |
| | 1 WEEK | $17.33_{\pm.08}$ | $\mathbf{17.28_{\pm.06}}$ | $\mathbf{17.33_{\pm.03}}$ | $17.40_{\pm.04}$ | $\mathbf{17.22_{\pm.06}}$ | $\mathbf{17.31_{\pm.07}}$ | $\mathbf{\color{red}17.11_{\pm.06}}$ |
| | 2 WEEKS | $17.13_{\pm.06}$ | $17.18_{\pm.14}$ | $17.15_{\pm.07}$ | $17.25_{\pm.03}$ | $17.15_{\pm.08}$ | $17.20_{\pm.09}$ | $\mathbf{\color{red}16.99_{\pm.05}}$ |

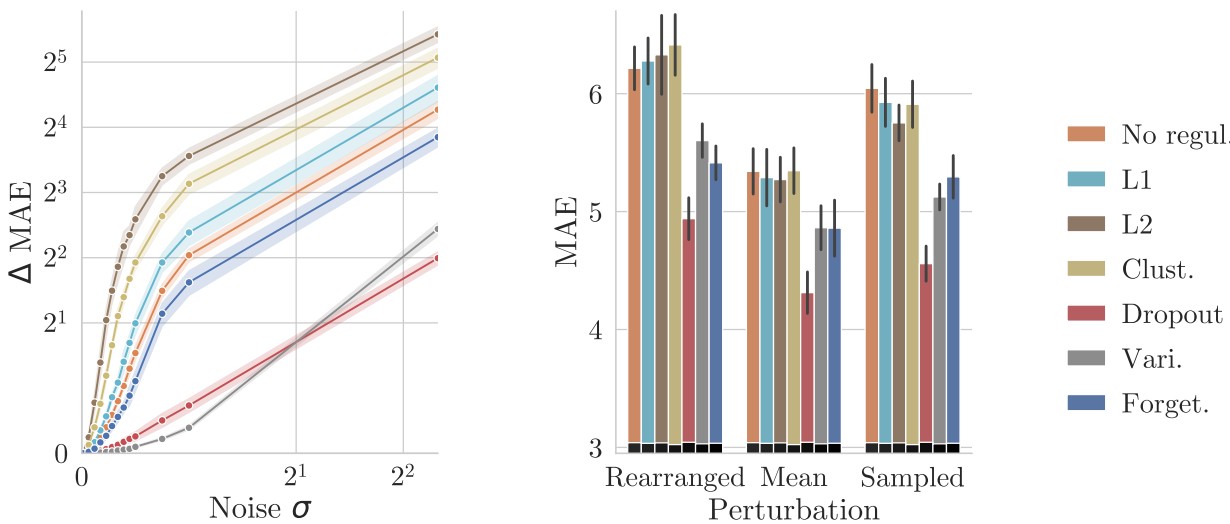

Figure 4: Test performance degradation on embeddings perturbation (5 runs, $\pm 1std$, STGNN, METR-LA). **Left:** Adding zero-mean Gaussian noise. **Right:** (**Left**) random shuffling, (**Middle**) replaced with their mean, and (**Right**) replaced by a draw from their sample normal.

### 5.4 Robustness to local parameter perturbation

Finally, we investigate how a model's forecasting accuracy is affected by the perturbation of the learned local embeddings. This provides insights into the robustness of the shared learned parameters and can serve as a proxy for the effectiveness of different regularizations in preventing co-adaptation. To avoid penalizing regularizations that are sensitive to the weights' magnitude (i.e., L2, L1), we consider perturbations that do not impact the scale of the learned representations. In particular, we experiment with four such strategies: adding zero-mean Gaussian noise to the embeddings (*Noise σ*), randomly shuffling embeddings across sequences (*Rearranged*), replacing each embedding with their sample mean (*Mean*), resampling each embedding from their sample normal distribution (*Sampled*) (see Appendix D.1 for a formal definition). Fig. 4 shows the results of the analysis after training an STGNN on the METR-LA dataset (complementary settings are illustrated in Appendix G). As one might expect, regularization methods that actively perturb the embeddings while learning (i.e., dropout, forgetting, variational) consistently result in more robust models under all the considered perturbations (**F2**) (**F3**). Conversely, other regularizations have marginal, or even negative, impact. We observe some consistency between the results observed in Fig. 4 and performance in transfer learning, shown in Tab. 3.

### 5.5 Discussion

Considering the observed empirical results, we can summarize a set of practical recommendations that can provide practitioners with a starting point for model search. In transductive settings, *variational* regularization performs reliably across benchmarks and it is a safe choice. Moreover, we suggest favoring techniques that perturb the embedding values during training, e.g., *variational* regularization, *forgetting* or *dropout*. In the context of transfer learning, our results suggest the adoption of *dropout*, when fine-tuning the model on a low amount of data, while *forgetting* or *variational* regularization can give better results as more data become available. The recommendation of prioritizing techniques that perturb the embeddings at training time remains valid in the transfer learning settings.

## 6 Conclusions

This paper highlights the importance of regularizing the learning of local embeddings in modern deep-learning architectures for time series forecasting. Our empirical study, across diverse datasets and scenarios, provides clear evidence that this practice is beneficial for a variety of reference architectures, representative of the state of the art. Notably, even simple techniques, when applied to local embeddings, can yield consistent performance gains. Furthermore, we observed that methods that actively perturb the embeddings at training time, such as dropout, *variational* regularization, and *forgetting*, consistently rank among the top performers, both in transductive and transfer learning settings. The use of this property as a guideline in designing new regularization methods is further supported by the effectiveness of the unprecedentedly considered *forgetting* regularization. Overall, our findings offer a strong empirical argument for the adoption of embedding regularization as standard practice when designing hybrid global-local architectures for time series. We believe this is an important building block towards developing more robust and transferable models for spatiotemporal data and designing foundation models for related time series processing.

**Limitations and future works**  While it is clear that these regularizations provide consistent advantages, it is difficult to strongly identify the best-performing method in the different scenarios. Hence, future works might focus on finding strategies that combine the best qualities of the different methods. Furthermore, future studies could try to provide methods to quantitatively and analytically characterize the co-adaptation of global and local parameters, for which we can only identify indirect signs. In particular, they might design experiments to detect the degeneration of local components into identifiers. Moreover, future research could explore how the efficacy of different regularization methods varies with the number of input time series and different downstream tasks.

**Acknowledgments**

Partly supported by International Partnership Program of the Chinese Academy of Sciences under Grant 104GJHZ2022013GC

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

## Appendix

## A  Datasets

Table 4: Dataset details.

| DATASETS | # TIME SERIES | TIME STEPS | CHANNELS | CONNECTIVITY | EDGES | SAMPLING RATE | TIME WINDOW | HORIZON |
|---|---|---|---|---|---|---|---|---|
| METR-LA | 207 | 34,272 | 1 | DIRECTED | 1515 | 5 MINUTES | 12 | 12 |
| PEMS-BAY | 325 | 52,128 | 1 | DIRECTED | 2369 | 5 MINUTES | 12 | 12 |
| CER-E | 485 | 25,728 | 1 | DIRECTED | 4365 | 30 MINUTES | 48 | 6 |
| AQI | 437 | 8,760 | 1 | UNDIRECTED | 2699 | 1 HOUR | 24 | 3 |
| CLM-D | 235 | 10,958 | 10 | NA | 2699 | 1 DAY | 14 | 3 |
| ENGRAD | 487 | 26,304 | 5 | NA | 2699 | 1 HOUR | 24 | 6 |
| PEMS03 | 358 | 26,208 | 1 | DIRECTED | 546 | 5 MINUTES | 12 | 12 |
| PEMS04 | 307 | 16,992 | 1 | DIRECTED | 340 | 5 MINUTES | 12 | 12 |
| PEMS07 | 883 | 28,224 | 1 | DIRECTED | 866 | 5 MINUTES | 12 | 12 |
| PEMS08 | 170 | 17,856 | 1 | DIRECTED | 277 | 5 MINUTES | 12 | 12 |

**METR-LA**  traffic data from Li et al. (2018). Traffic readings are from different loop detectors on highways in the Los Angeles County. Licensed under Attribution 4.0 International (CC BY 4.0).

**PEMS-BAY**  traffic data from Li et al. (2018). Traffic readings are from different loop detectors on highways in the Los Angeles County. Licensed under Attribution 4.0 International (CC BY 4.0).

**CER-E**  Electric load data from (CER, 2016). Data encompass energy consumption readings from smart meters in small and medium enterprises, collected in the context of the Commission for Energy Regulation (CER) Smart Metering Project. Data access can be requested through `https://www.ucd.ie/issda/data/commissionforenergyregulationcer/`

**AQI**  Air quality data from Zheng et al. (2015). The dataset collects measurements of the *PM2.5* pollutant from air quality stations in 43 Chinese cities and is available at `https://www.microsoft.com/en-us/research/publication/forecasting-fine-grained-air-quality-based-on-big-data/`

**CLM-D**  satellite daily climatic dataset from (De Felice et al., 2024). Data were obtained from the POWER Project's Daily 2.3.5 version on 2023/02/26 and sampled in the correspondence of the 235 world capitals. Further information, together with data and API, is available at the project website (`https://power.larc.nasa.gov/`). For daily data, we select the following 10 variables: *mean temperature* ($°C$), *temperature range* ($°C$), *maximum temperature* ($°C$), *wind speed* ($m/s$), *relative humidity* (%), *precipitation* ($mm$/day), *dew/frost point* ($°C$), *cloud amount* (%), *allsky surface shortwave irradiance* ($W/m^2$) *and all-sky surface longwave irradiance* ($W/m^2$). Data extend for 30 years (1991 to 2022).

**EngRAD**  hourly climatic dataset from (Marisca et al., 2024). The measurements are provided by `https://open-meteo.com` (Zippenfenig, 2023) and licensed under Attribution 4.0 International (CC BY 4.0). Data are collected on a grid in correspondence with cities in England. The variables correspond to *air temperature* at 2 meters above ground ($°C$); *relative humidity* at 2 meters above ground (%); summation of total *precipitation* (rain, showers, snow) during the preceding hour ($mm$); total cloud cover (%); global horizontal irradiation ($W/m^2$). Data extend for 3 years (2018 to 2020).

**PEMS03, PEMS04, PEMS07, and PEMS08**  datasets from Guo et al. (2021) collect traffic detector data from 4 districts in California provided by Caltrans Performance Measurement System (PeMS). Data are aggregated into 5-minutes intervals.

All datasets were split *70%/10%/20%* into *train*, *validation* and *test* along the temporal axis. Datasets with a high number of missing values, i.e., METR-LA, PEMS-BAY and AQI, have a binary mask concatenated to the input, indicating if the corresponding value has been artificially imputed. Note that, unless differently

specified, used datasets are public domain. Where possible, we pointed to the original source of the data or to a meaningful reference.

## B  Models

In this section we describe the models used in our study. Note that all models used a fixed hidden size $d_h$ for all layers. For all the architectures, the ENCODER 3 is parametrized by a linear layer, while the DECODER is a 1-layer MLP followed by $H$ parallel linear layers, each decoding a different step in the forecasting horizon. The RNN model is implemented by means of a 1-layer GRU (Cho et al., 2014b) shared among all sequences. On top of the same GRU architecture, for the STGNN model, we employ 2 layers of message passing defined as

$$\mathbf{m}^{j\to i} = \mathbf{W}_2 \xi \left( \mathbf{W}_1 \left[ \boldsymbol{h}^i, \boldsymbol{h}^j, a_{ji} \right] \right), \qquad \alpha^{j\to i} = \eta \left( \mathbf{W}_0 \mathbf{m}^{j\to i} \right), \tag{8}$$

$$\tilde{\mathbf{h}}^i = \xi \left( \mathbf{W}_3 \boldsymbol{h}^i + \sum_{j\in\mathcal{N}(i)} \{ \alpha^{j\to i} \odot \mathbf{m}^{j\to i} \} \right), \tag{9}$$

where $\mathbf{W}_0 \in \mathbb{R}^{d_h \times d_h}$, $\mathbf{W}_1 \in \mathbb{R}^{d_h \times (2d_h+1)}$, $\mathbf{W}_2 \in \mathbb{R}^{d_h \times d_h}$ and $\mathbf{W}_3 \in \mathbb{R}^{d_h \times d_h}$ are learnable parameters, $[\cdot,\cdot]$ is the concatenation operator along the feature dimension, $\odot$ is the Hadamard product, $\xi$ denotes the *elu* (Clevert et al., 2016) activation function and $\eta$ the sigmoid activation function. Furthermore, $\mathcal{N}(i)$ denotes the neighbours of the $i$-th sequence, induced by the adjacency matrix $\mathbf{A}$, $a_{ji}$ denotes the weight associated to edge $j \to i$ and $\boldsymbol{h}^i$ and $\boldsymbol{h}^j$ denote the hidden features associated with the $i$-th and $j$-th sequences respectively. Regarding the STAtt model, we use 2 layers of multi-head attention (Vaswani et al., 2017) taking as input tokens the hidden representations extracted by a temporal encoder which shares the same architecture as the aforementioned RNN. Finally, for consistency with the original experiment (Cini et al., 2023b), we employ an STGNN consisting of a 1-layer GRU and 2 message passing layer implementing

$$\tilde{\mathbf{h}}^i = \xi \left( \mathbf{W}_4 \boldsymbol{h}^i + \frac{1}{\|\mathcal{N}(i)\|} \sum_{j\in\mathcal{N}(i)} \{ \mathbf{W}_5 \boldsymbol{h}^j \} \right), \tag{10}$$

where $\mathbf{W}_4 \in \mathbb{R}^{d_h \times d_h}$ and $\mathbf{W}_5 \in \mathbb{R}^{d_h \times d_h}$ are learnable parameters, while $\|\cdot\|$ is the cardinality operator.

## C  Additional details on forgetting regularization

For the sake of completeness, in the following, we list some additional details regarding the local forgetting regularization.

### C.1  Embedding-related Encoder/Decoder parameters reset

When resetting the local parameters, we also reset the encoder's (Eq. 3) and decoder's (Eq. 5) parameters (i.e., coefficients of a linear layer) that directly interact with the embeddings' features. As an illustrative example, consider an ENCODER parametrized by a MLP, with input linear layer

$$\boldsymbol{h}_t^0 = [\boldsymbol{X}_{t-1} \| \boldsymbol{U}_{t-1} \| \boldsymbol{E}] \, \boldsymbol{W}^T + \boldsymbol{b}, \tag{11}$$

In this case, the encoder's parameters to be reset would correspond to the last $d_e$ columns of the weight matrix $\boldsymbol{W}$. An equivalent behavior is implemented for the decoder's input layer.

### C.2  Forgetting period sensitivity

The introduced *forgetting* regularization has two hyperparameters: the reset period $k$ and the halting epoch. While the latter can be determined automatically, by monitoring the validation error before each reset, the former should be set empirically. To provide insight on the degree to which the selection of $k$ can impact

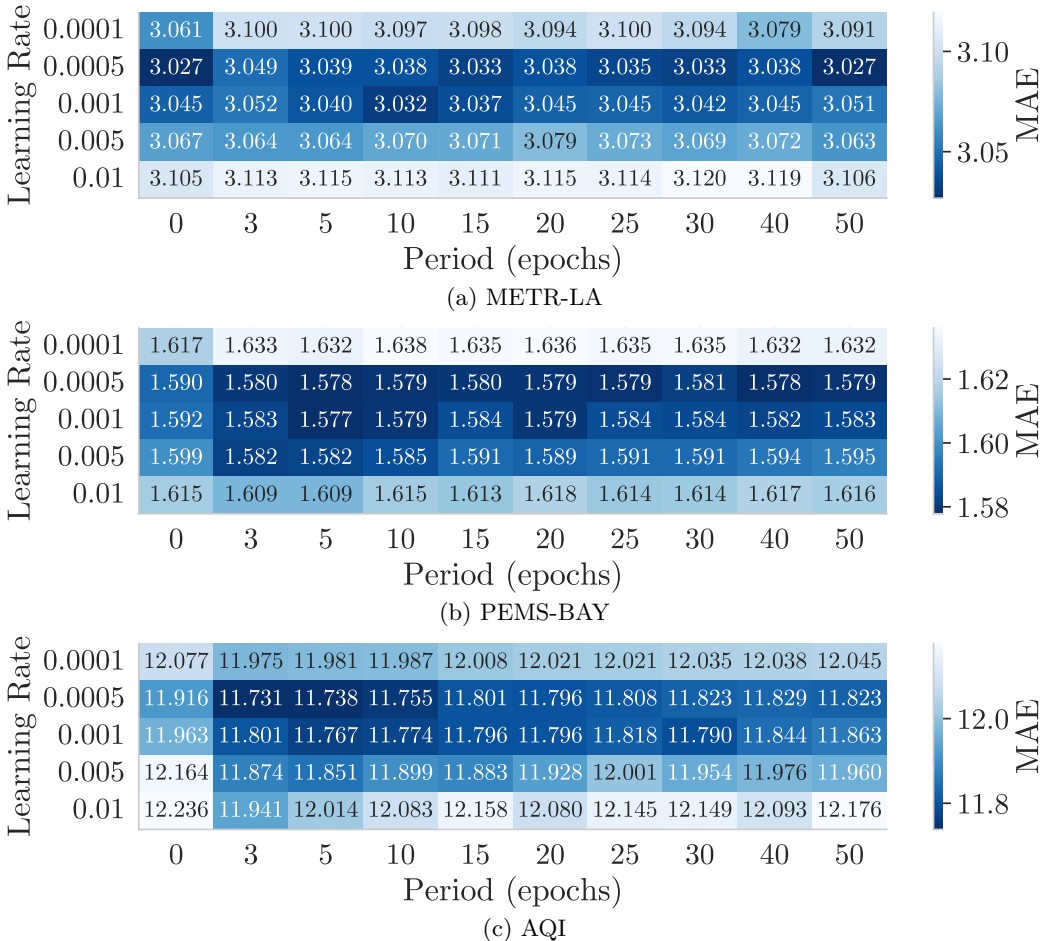

Figure 5: Sensitivity to forgetting period $k$ across different learning rates and datasets (5 runs, STGNN). A period of 0 indicates no regularization.

regularization performance, we perform a sensitivity study on 3 datasets: METR-LA, PEMS-BAY and AQI. The results are shown in Fig. 5, where we report the mean absolute error (MAE) with varying learning rates and values of $k$. We can see that performance is relatively stable in regard to changes in $k$, suggesting that hyperparameter search can be limited to few values (e.g., 3 for a short period and 30 for a long one). Another possibility, to avoid searches, is to select the value by hand, after inspecting the learning curves of the unregularized model, in order to select a reset period that allows performance recovery.

### C.3 Forgetting warm-up

To avoid instabilities at the beginning of training, particularly when using short reset periods, the forgetting routine can be initiated after a warm-up period, which can be set empirically to a few epochs, as commonly done with learning rate warm-up procedures or with other regularizations (Cini et al., 2023b).

## D Detailed experimental setting

**Reproducibility** Python code to reproduce the experiments is available online[1]. Datasets will be automatically downloaded or access can be requested to the authors, aside from CER-E and EngRAD. Such datasets can be obtained by contacting the original papers' authors (EngRAD (Marisca et al., 2024)) or by requesting

---

[1]https://github.com/LucaButera/TS-embedding-regularization

data access to the appropriate authority (CER-E (CER, 2016)). To specifically reproduce the experiments in Tab 3, regarding the columns for *clustering* and *variational* regularizations, access to the code can be requested to the original authors of Cini et al. (2023b). Appendix A contains pointers to request such access. Note that randomized operations in the code are controlled by fixed seeds for reproducibility purposes.

**Shared settings** All the models were trained with the Adam optimizer (Kingma & Ba, 2015), with a batch size of 64 and up to 300 batches per epoch. We used the *Python* (Van Rossum & Drake, 2009) programming language, leveraging *Torch Spatiotemporal* (Cini & Marisca, 2022), *Pytorch* (Paszke et al., 2019) and *Pytorch Lightning* (Falcon & The PyTorch Lightning team, 2019) to implement all the experiments. Experiments were scheduled and logged by leveraging *Hydra* (Yadan, 2019) and *Weights and Biases* (Biewald, 2020). The mean absolute error (MAE) was used as loss function in all experiments. For *variational* and *clustering* regularizations additional hyperparameters, aside from the regularization strength, i.e., $\lambda_{var}$, $\lambda_{clst}$, were taken from the original paper. Unless specified, we employed and embedding size $d_e = 32$ for all models. Moreover, all datasets used temporal encodings as additional covariates, in particular, a one-hot encoding of the weekday and sinus and cosinus encodings with daily period.

**Experiment specific hyperparameters** For the experiments in Tab. 1, optimal hyperparameters, i.e., learning rate $lr$ and hidden size $d_h$, for each model and dataset, were found via a grid-search over $lr \in [0.00025, 0.00075, 0.0015, 0.003]$ and $d_h \in [32, 64, 128, 256]$, with constraints caused by GPU memory capacity in some settings. Before the grid search for regularized models, regularization's hyperparameters $\lambda_{l1}$, $\lambda_{l2}$, $\lambda_{var}$ and $\lambda_{clst}$, were set according to the following procedure. For each model and dataset (we here consider three datasets: METR-LA, PEMS-BAY and AQI), the validation error is computed for regularization strength values in $[0.01, 0.001, 0.0001, 0.00001]$, while keeping the $lr$ and $d_h$ fixed to the optimal values for the respective un-regularized model. For each regularization hyperparameter, the corresponding validation results are ranked and the average rank is computed across models and datasets. The parameter with the top rank is selected and kept fixed. This resulted in: weight of L2 $\lambda_{l2} = 0.0001$, weight of L1 $\lambda_{l1} = 0.00001$, weight of *variational* regularization $\lambda_{var} = 0.00005$ and weight of *clustering* regularization $\lambda_{clst} = 0.0005$. Dropout's probability of dropping a connection was set to $p = 0.5$, which is a middle ground between soft and aggressive dropout. Moreover, we set the resetting period of *forgetting* to $k = 20$ epochs with 30 epochs of warm-up. The analysis in Sec. C.2 shows how the method is not particularly sensitive in this regard. Forgetting was halted after 150 epochs. Each model was trained for 200 epochs and the best validation parameters were used for testing.

For the experiments in Tab. 3, we used the STGNN in Eq. 10 and set $d_h = 64$, $\lambda_{var} = 0.05$ and $\lambda_{clst} = 0.5$ as in Cini et al. (2023b), while for the other regularizations we used the same hyperparameters as in Tab. 1. Furthermore, we used a learning rate $lr = 0.005$ during training and $lr = 0.001$ during fine-tuning. The training lasted up to 150 epochs with 50 epochs of early stopping patience, while fine-tuning lasted up to 1000 epochs with 100 epochs of patience. No temporal encodings were used in this setting.

For experiments in Sec. 5.2 and Sec. 5.4 we used set $d_h = 64$ and $lr = 0.00075$. Regularization's hyperparameters were set to the values found for Tab. 1. Training lasted up to 300 epochs, early stopping patience was set to 100, while forgetting was halted after 100 epochs. Note that the *forgetting* regularization ignores early stopping for the whole duration before being halted; this is to avoid triggering early stopping as a byproduct of the parameter resetting itself.

**Metrics** For univariate datasets, we consider the mean absolute error (MAE), as this is a standard choice across the related literature. For multivariate datasets, i.e., CLM-D and EngRAD, following De Felice et al. (2024), we compute the mean relative error (MRE) independently for each channel, then, take the channel-wise average to obtain a unique final score, which we termed multivariate mean relative error (MMRE) in Tab. 1.

**Computing resources** Experiments were run on A100 and A5000 NVIDIA GPUs. The vast majority of the experiments conducted in our work can be easily run on moderate GPU hardware, with at least 8 GBs of VRAM. However, not that some experiments, may require more. For instance, the transfer experiments require at least 20 GBs of VRAM on some of the benchmark dataset. Moreover, high hidden size configurations of STGNN and STAtt models, required up to 40 GBs of VRAM. In general, a single run (i.e., training one

model on one dataset), given hardware that fulfills the memory requirements, takes from 30 minutes to 3 hours, depending on the specific configuration.

### D.1 Embeddings perturbation

Here we provide a more formal definition of the perturbations adopted in Sec. 5.4. In particular, considering $N$ embedding vectors:

- *Noise $\sigma$* consists in adding zero-mean gaussian noise, from an isotropic multivariate normal distribution, to the embeddings. Formally, this refers to substituting each embedding $\mathbf{e}^i$ with $\mathbf{e}^i + \epsilon$, where $\epsilon \sim \mathcal{N}\left(\mathbf{0}, diag\left(\sigma^2\right)\right)$.

- *Rearrange* refers to randomly reassigning the embedding vectors to different time-series in the collection. Formally, this means substituting each embedding $\mathbf{e}^i$ with an embedding $\mathbf{e}^j$, where $j \sim Multinomial\left(\{0, ..., N-1\}\right)$ is a sample from a multinomial distribution over the embedding indices, sampled without repetition.

- *Mean* refers to setting each embedding to the mean values across embeddings themselves. Formally, replacing each embedding $\mathbf{e}^i$ with $\frac{1}{N}\sum_{j=0}^{N}\mathbf{e}^j$.

- *Sampled* consists in estimating the sample gaussian distribution of the embeddings, and then replacing each embedding with a draw from such distribution. Formally, this entails estimating the gaussian's parameters as $\mu_e = \frac{1}{N}\sum_{i=0}^{N-1}\mathbf{e}^i$ and $\sigma_e^2 = \frac{1}{N-1}\sum_{i=0}^{N-1}\left(\mathbf{e}^i - \mu\right)$. Then we sample values for each embedding as $\mathbf{e}^i \sim \mathcal{N}\left(\mu_e, \sigma_e^2\right)$.

Note that, to avoid penalizing regularizations that are sensible to the weights' magnitude (i.e., L1, L2), we consider perturbations that do not change the scale of the learned representations. As this does not hold true for the *Noise $\sigma$* perturbation, since it can potentially affect the embeddings magnitude, depending on the standard deviation, we selected most values to be within reason relative to the embeddings magnitude itself. Nonetheless, we also employed a few more extreme values, in order to observe their effects.

## E Effect on simple global models

The purpose of model regularization is, usually, to limit model capacity by means of additional constraints and, in turn, obtain models that generalize better. This implies that excessive regularization might result in a degradation of performance. In principle, regularization of local parameters should not have a negative impact on underparametrized global models, as their parameters are not constrained directly. To evaluate this, we employ a simple 2-layer MLP as our global model (Eq. 4), and train it with the same settings as in Sec. 5.1.

Tab. 5 reports the obtained results in case of tuned learning rate and hidden size of 64 and 128 units. Compared to models in Tab. 1, the performance of the MLP is worse or comparable at best, however, the regularizations relative effectiveness is similar, considering each dataset, at hidden size 128. In the extreme case of the MLP with hidden size 64, we can observe additional scenarios in which most regularizations are ineffective (i.e., CER-E). Nonetheless, we do not observe catastrophic effects on performance. Notably, with some datasets, i.e., AQI and EngRAD, adding local parameters to the global MLP model hurts performance, while regularization allows an effective exploitation of the embeddings.

## F Effect of combining different regularizations

To further investigate whether different regularization techniques can be combined for improved performance, we applied L2 and dropout on top of the approaches specifically designed for the regularization of local embeddings, i.e., *clustering*, *variational* and *forgetting* regularizations. This covers the most reasonable combinations, as L2 and *dropout* are commonly used in combination with other regularization approaches. In particular, we consider the same transductive setting as in Sec. 5.1, and adopt the optimal hyperparameter

Table 5: Forecasting error in transductive setting for MLP model (hidden size 64-128) (5 runs, $\pm 1 std$). Equal to/better than unregularized (+ EMB) in bold. Best in red. '+ REGULARIZATION' denotes the addition of that specific regularization only, on top of '+ EMB.'.

| DATASET | METR-LA | PEMS-BAY | CER-E | AQI | CLM-D | ENGRAD |
|---|---|---|---|---|---|---|
| MODEL | MAE ↓ | MAE ↓ | MAE ↓ | MAE ↓ | MMRE ↓ | MMRE ↓ |
| MLP (64) | $3.580_{\pm0.005}$ | $1.808_{\pm0.002}$ | $0.4658_{\pm0.0008}$ | $\mathbf{13.398}_{\pm0.044}$ | $19.86_{\pm0.01}$ | $\mathbf{31.38}_{\pm0.13}$ |
| + EMB. | $3.153_{\pm0.012}$ | $1.619_{\pm0.005}$ | $0.4233_{\pm0.0007}$ | $13.521_{\pm0.082}$ | $19.55_{\pm0.01}$ | $31.40_{\pm0.14}$ |
| + L1 | $\mathbf{3.150}_{\pm0.005}$ | $\mathbf{1.614}_{\pm0.006}$ | $0.4236_{\pm0.0008}$ | $\mathbf{13.392}_{\pm0.067}$ | $19.56_{\pm0.01}$ | $\mathbf{31.32}_{\pm0.04}$ |
| + L2 | $\mathbf{3.147}_{\pm0.007}$ | $\mathbf{1.613}_{\pm0.004}$ | $0.4242_{\pm0.0010}$ | $\mathbf{13.389}_{\pm0.042}$ | $19.57_{\pm0.02}$ | $\mathbf{31.21}_{\pm0.09}$ |
| + CLUST. | $\mathbf{3.152}_{\pm0.014}$ | $1.625_{\pm0.009}$ | $0.4236_{\pm0.0012}$ | $13.478_{\pm0.070}$ | $\color{red}19.54_{\pm0.01}$ | $\mathbf{31.25}_{\pm0.14}$ |
| + DROP. | $3.193_{\pm0.003}$ | $1.630_{\pm0.006}$ | $0.4317_{\pm0.0009}$ | $\color{red}13.328_{\pm0.036}$ | $19.62_{\pm0.01}$ | $\color{red}31.20_{\pm0.07}$ |
| + VARI. | $\color{red}3.139_{\pm0.011}$ | $\color{red}1.610_{\pm0.001}$ | $\color{red}0.4216_{\pm0.0011}$ | $\mathbf{13.337}_{\pm0.020}$ | $\color{red}19.54_{\pm0.01}$ | $\mathbf{31.32}_{\pm0.03}$ |
| + FORG. | $3.157_{\pm0.013}$ | $\mathbf{1.617}_{\pm0.007}$ | $0.4271_{\pm0.0012}$ | $\mathbf{13.472}_{\pm0.099}$ | $19.56_{\pm0.02}$ | $\mathbf{31.34}_{\pm0.09}$ |
| MLP (128) | $3.571_{\pm0.002}$ | $1.798_{\pm0.003}$ | $0.4557_{\pm0.0003}$ | $\mathbf{13.390}_{\pm0.044}$ | $19.78_{\pm0.01}$ | $31.44_{\pm0.04}$ |
| + EMB. | $3.157_{\pm0.014}$ | $1.615_{\pm0.006}$ | $0.4141_{\pm0.0013}$ | $13.534_{\pm0.052}$ | $\color{red}19.47_{\pm0.01}$ | $31.38_{\pm0.10}$ |
| + L1 | $\mathbf{3.152}_{\pm0.004}$ | $\mathbf{1.604}_{\pm0.006}$ | $\mathbf{0.4128}_{\pm0.0007}$ | $\mathbf{13.417}_{\pm0.078}$ | $19.49_{\pm0.02}$ | $\mathbf{31.38}_{\pm0.07}$ |
| + L2 | $\mathbf{3.150}_{\pm0.015}$ | $\color{red}1.600_{\pm0.006}$ | $\mathbf{0.4117}_{\pm0.0009}$ | $\mathbf{13.450}_{\pm0.059}$ | $19.48_{\pm0.01}$ | $31.65_{\pm0.17}$ |
| + CLUST. | $\mathbf{3.143}_{\pm0.017}$ | $\mathbf{1.603}_{\pm0.003}$ | $\mathbf{0.4134}_{\pm0.0009}$ | $13.612_{\pm0.119}$ | $\color{red}19.47_{\pm0.01}$ | $31.42_{\pm0.06}$ |
| + DROP. | $3.195_{\pm0.008}$ | $1.624_{\pm0.003}$ | $\color{red}0.4101_{\pm0.0003}$ | $\color{red}13.320_{\pm0.044}$ | $19.53_{\pm0.02}$ | $\mathbf{31.35}_{\pm0.13}$ |
| + VARI. | $\color{red}3.139_{\pm0.017}$ | $\mathbf{1.603}_{\pm0.005}$ | $\mathbf{0.4109}_{\pm0.0012}$ | $\mathbf{13.346}_{\pm0.038}$ | $\color{red}19.47_{\pm0.01}$ | $\mathbf{31.33}_{\pm0.07}$ |
| + FORG. | $\mathbf{3.150}_{\pm0.014}$ | $1.616_{\pm0.010}$ | $0.4184_{\pm0.0008}$ | $\mathbf{13.427}_{\pm0.058}$ | $19.53_{\pm0.02}$ | $\mathbf{31.36}_{\pm0.09}$ |

Table 6: Forecasting test error under optimal model size and learning rate from Tab. 1 (STGNN, 5 runs, $\pm 1 std$) when adding additional regularizations. Combinations equal to or better than the corresponding base regularization are in bold. The best-performing method within each dataset and base regularization is in red. Regularization methods are always considered applied on top of the '+ EMB.' model.

| DATASET | METR-LA | PEMS-BAY | AQI |
|---|---|---|---|
| MODEL | MAE ↓ | MAE ↓ | MAE ↓ |
| + CLUST. | $3.025_{\pm.012}$ | $1.580_{\pm.005}$ | $11.876_{\pm.053}$ |
| + CLUST. + L2 | $\color{red}3.019_{\pm0.008}$ | $1.581_{\pm0.008}$ | $\mathbf{11.767}_{\pm0.023}$ |
| + CLUST. + DROP. | $3.052_{\pm0.007}$ | $\color{red}1.572_{\pm0.009}$ | $\color{red}11.712_{\pm0.037}$ |
| + VARI. | $\mathbf{3.013}_{\pm.005}$ | $\mathbf{1.566}_{\pm.003}$ | $11.768_{\pm.026}$ |
| + VARI. + L2 | $3.037_{\pm0.009}$ | $1.581_{\pm0.007}$ | $11.789_{\pm0.022}$ |
| + VARI. + DROP. | $3.047_{\pm0.008}$ | $1.576_{\pm0.004}$ | $\color{red}11.675_{\pm0.023}$ |
| + FORG. | $3.050_{\pm.017}$ | $1.578_{\pm.006}$ | $11.793_{\pm.040}$ |
| + FORG. + L2 | $\mathbf{3.045}_{\pm0.012}$ | $\mathbf{1.576}_{\pm0.007}$ | $\mathbf{11.764}_{\pm0.023}$ |
| + FORG. + DROP. | $3.087_{\pm0.005}$ | $\color{red}1.570_{\pm0.003}$ | $\color{red}11.755_{\pm0.028}$ |

configuration (see Appendix D for details) previously found for *clustering*, *variational*, and *forgetting*, respectively.

Tab. 6 shows the obtained results. We can see that combining different regularization techniques can be beneficial. Noticeably, combining *variational* regularization and dropout yielded a new overall best score on the AQI dataset. However, improvements appear mostly case-dependent, without emerging clear patterns.

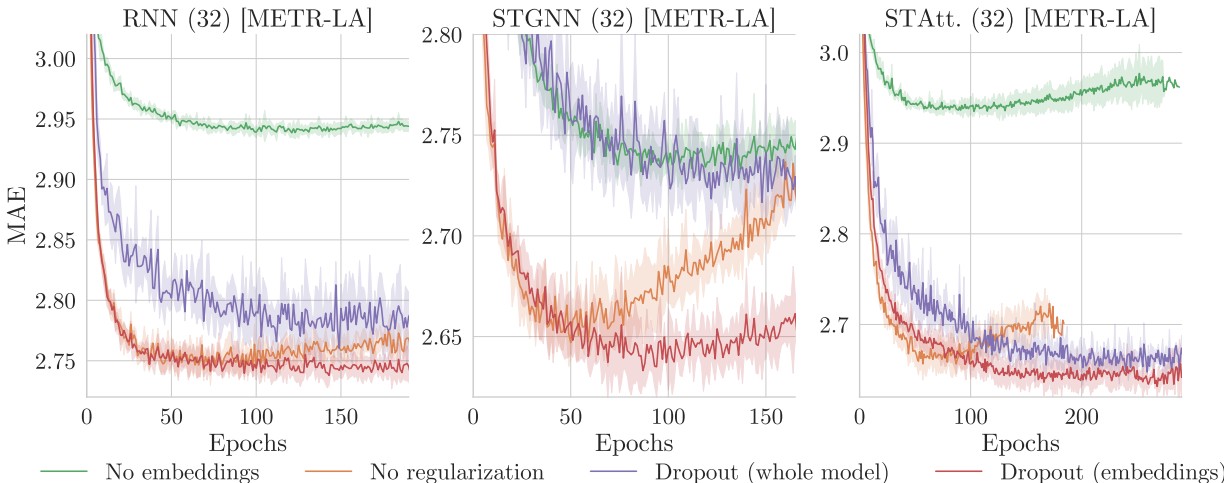

Figure 6: Validation curves for different model families (5 runs, $\pm 1 std$). Curves show the application of dropout regularization at the embedding level or over the entire architecture. Plot names follow the convention *model (embedding size) [dataset]*.

## G  Additional experiments

For completeness, in this section, we provide experimental results in addition to what has been shown in the main paper.

Fig. 2 shows an example of how regularizing local parameters only can have a different impact than regularizing all parameters on the training of an STGNN model. Fig. 6 complements what has been shown in such an example by providing results for the other two models, i.e., RNN and STAtt. A similar pattern emerges for all the considered models, though with different degrees.

Similarly, Fig. 7 complements the results shown in Fig. 3, with the models that were not shown. We can notice a similar pattern in which dropout (*red*) can be problematic at smaller embedding sizes (top row). Nonetheless, this seems to be less significant for the STGNN in this specific scenario. In general, the additional plots confirm that dropout (*red*) and forgetting (*blue*) lead to different learning curves compared to the un-regularized model (*orange*), while other regularizations have little impact on this aspect.

Fig. 8 shows results obtained by training models with the same setting as Sec. 5.2 on the PEMS-BAY dataset. We can see that, in this case, the validation curves barely plateau. In this scenario, dropout shows potentially problematic effects, similar to those observed for small embedding sizes.

Finally, Fig. 9 and Fig. 10, provide complementary results for Fig. 4. We can notice how the different regularizations rank similarly, in terms of global model robustness, across different architectures.

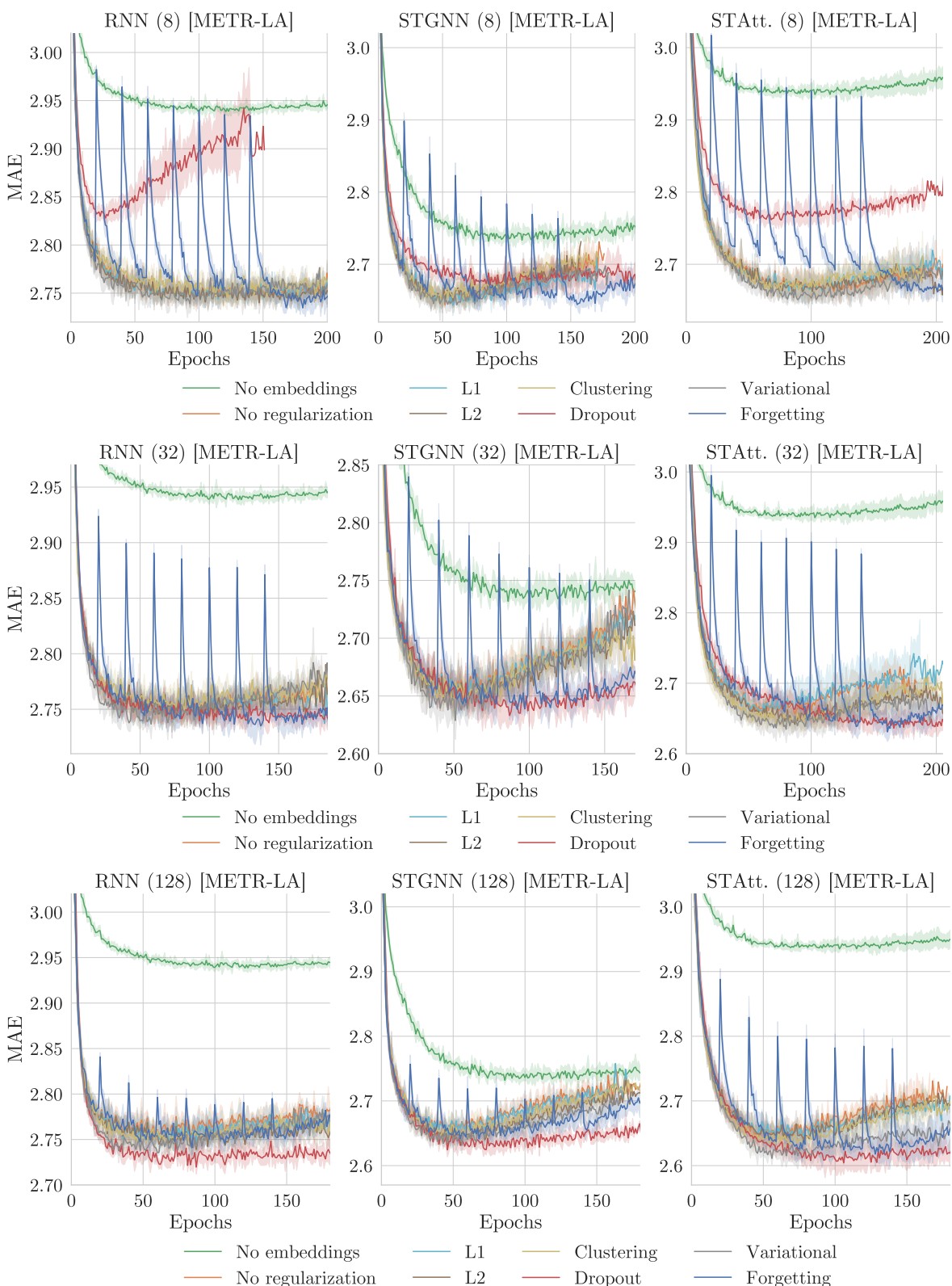

Figure 7: Validation curves for different model families (5 runs, ±1*std*). Plot names follow the convention *model (embedding size) [dataset]*.

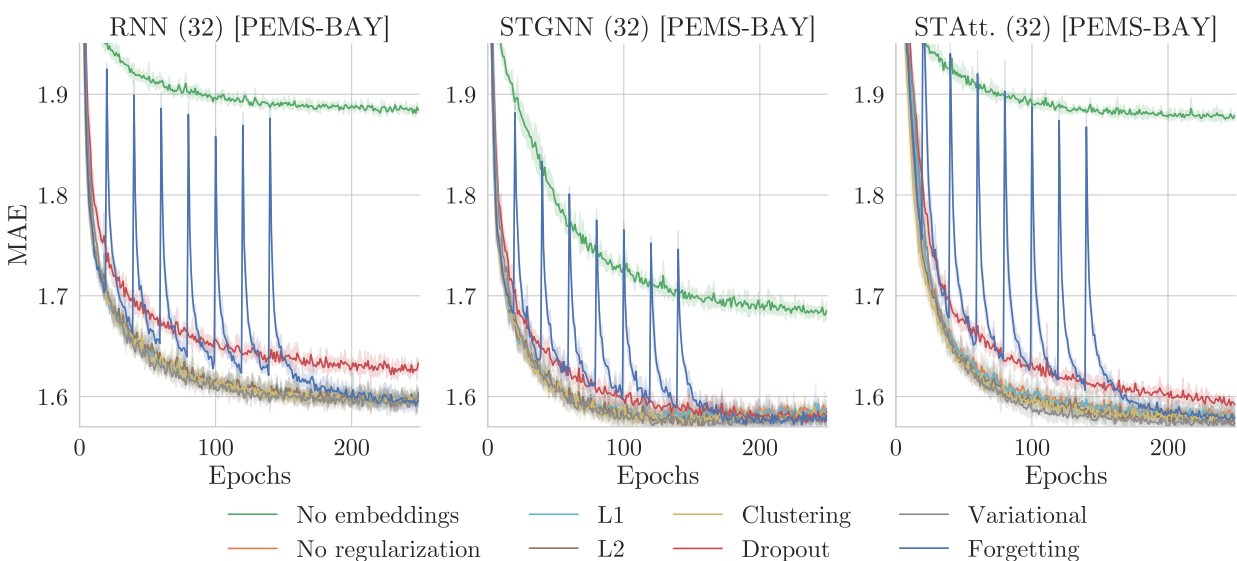

Figure 8: Validation curves for different model families (5 runs, $\pm 1 std$). Plot names follow the convention *model (embedding size) [dataset]*.

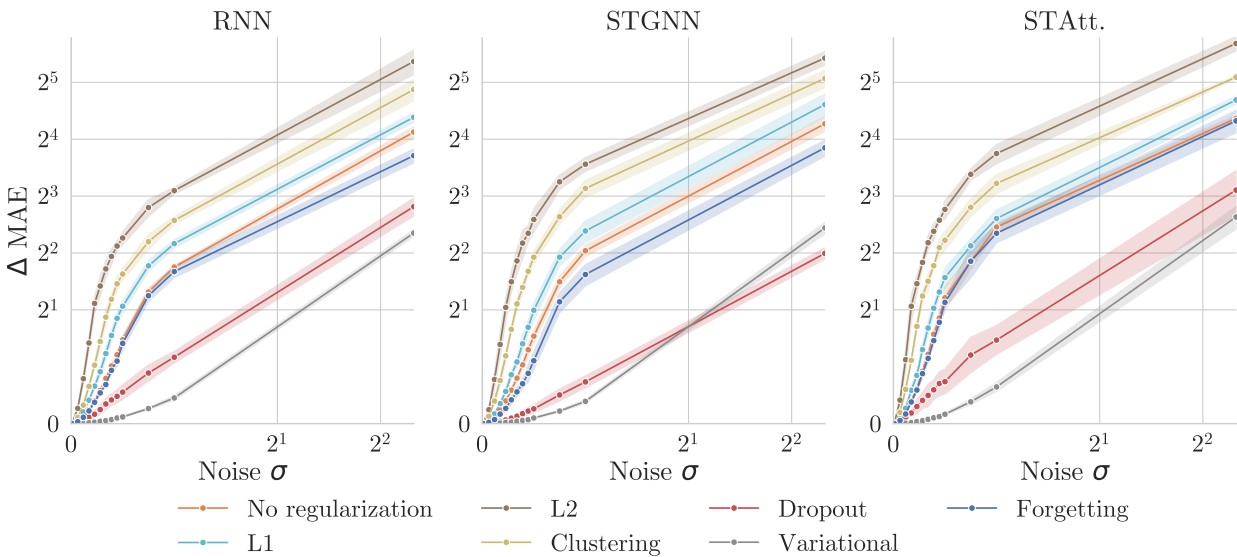

Figure 9: Test performance degradation when adding zero-mean Gaussian noise to the embeddings, with increasing variance (5 runs, $\pm 1 std$, METR-LA). Fixed hidden size and learning rate, 64 and 0.00075 respectively.

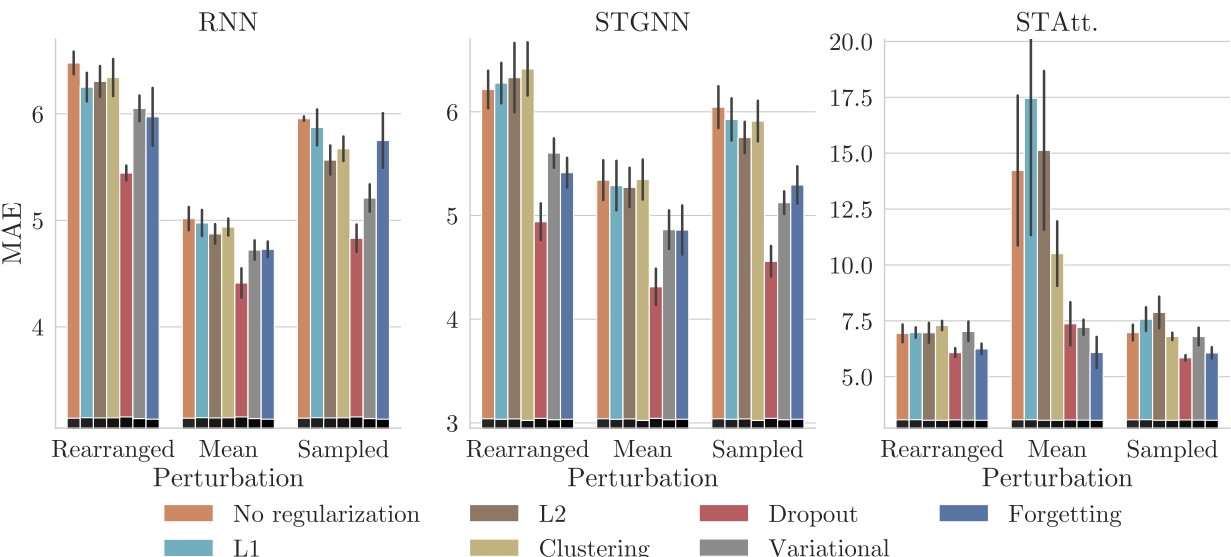

Figure 10: Test performance degradation on embeddings perturbation (5 runs, $\pm 1 std$, METR-LA).(**Left**) random shuffling, (**Middle**) replaced with their mean, and (**Right**) replaced by a draw from their sample normal. Fixed hidden size and learning rate, 64 and 0.00075 respectively.

