# OpenReview forum: "On the Regularization of Learnable Embeddings for Time Series Forecasting"
_TMLR — Accepted by TMLR_

### Review · Reviewer_LgE4 · 2024-11-08

**Summary Of Contributions:**

Given a collection of time series, this paper considers forecasting in two settings:
(1) Forecasting in an unknown time series with little to no information on the new time series during training time.
(2) Forecasting in different realizations of the known time series.
When processing sets of time series, one can learn a single (global) model processing all time series equally, an individual model for each time series, or a combination of both. This paper considers a general architecture combining a local encoder and a local decoder with a global model in between. The main contribution of the paper is an experimental comparison of multiple regularization techniques applied to the local parts of the combined model in both settings.

**Audience:**

Yes

**Claims And Evidence:**

Yes

**Requested Changes:**

The language in the summary of the findings seems too strong. "clear improvements", "larger performance gains", and "significant performance improvements" seem to imply much stronger results than the experiments seem to show. A more conservative wording might be more appropriate.

The paper spends a lot of time on the definition of the models in varying generality. That space might be better used to expand the discussion of the results, the conclusion, and the discussion of the regularization techniques, in particular the proposed method.

It is not clear, how the "simple setting" in Figure 2 relates to the benchmarks. Is there a strong reason, why global regularization should not also be considered in all evaluations as a baseline?

The paper would greatly benefit from some general improvements on the writing.
Here are some issues i noticed:
- P2§2 "[...] that operate on fixed size dictionary [...]" -> "[...] that operate on a fixed size dictionary [...]"
- "w.r.t." is usually more appropriate for theoretical papers.
- Equation (3) seems to be missing W
- Section 3 §1 "[...] and, often, as spatial positional encodings [...]" -> "[...] and, often as spatial positional encodings [...]"
- Section 4.1 §2 "L1 and L2 regulariation" -> "L1 and L2 regularization
- "embedding" is usually used for representations of data.
- $\sigma$ is used in multiple different contexts.
- Caption of Table 3 "[...]  methods equal or better [...]" -> "[...]  methods equal to or better [...]"
- Caption of Table 5 "trasductive" -> "transductive"
- Using $\\Vert$ for concatenation seems a bit confusing at first, especially when used in close proximity to the norm, as e.g. in (8) and (10).
- There seems to be a problem with the definition of $W_0, \\ldots, W_3$ or equation (8-9). $W_1$ and $W_3$ don't seem to come from the same space.
- Sec 2.2 §1 "With full generality [...]" seems to be contradicting the following statement, where only a special case is considered.
- Sec 2.2 §3 The indices of the parameters should probably also start at 1.
- $<t$ is usually used to refer to values less than $t$. Using $\leq t$ might make the equations more intuitive.

**Strengths And Weaknesses:**

## Strengths
The paper shows, that regularizing the local components yields better performance in both settings.
In particular, the paper shows that Dropout is most effective in the first setting, if less information is available.

## Weaknesses
The writing could use some work.

Especially in the second setting, the claimed "clear performance improvements" seem to be very small for most models. The conclusion of the authors seem to imply much stronger results.

Even after looking at the appendix, it is not entirely clear, which parts of which models were tuned and which parts were taken from related work, making it difficult to fully contextualize the presented results.

I am not certain about the novelty of the proposed "forgetting regularization". After a quick look into the cited papers, i fail to see significant differences to prior work and the paper does not provide a satisfying comparison.

---

> ### Author Response · Authors · 2024-11-21
>
> We thank the reviewer for their comments. In the following, we address the mentioned weaknesses and requested changes.
>
> >**Weakness 1 and Requested change 1** Especially in the second setting, the claimed "clear performance improvements" seem to be very small for most models. The conclusion of the authors seem to imply much stronger results. [...] The language in the summary of the findings seems too strong. "clear improvements", "larger performance gains", and "significant performance improvements" seem to imply much stronger results than the experiments seem to show. A more conservative wording might be more appropriate.
>
> While we agree that the performance gains are not always impressive, we think they are consistent across settings and cannot be disputed. Furthermore, all of the methods are very easy to implement and bear little to no overhead. We agree with the reviewer that the presentation can be improved in this regard and we will rephrase the relevant paragraphs accordingly to better convey the actual observed results. The revision will be uploaded shortly.
>
> >**Weakness 2** Even after looking at the appendix, it is not entirely clear, which parts of which models were tuned and which parts were taken from related work, making it difficult to fully contextualize the presented results.
>
> Are you referring to hyperparameters and specific layers? We use reference architectures adapted from the framework proposed by [1]. The structure of the different architectures is mentioned in Appendix B and it is fixed across experiments. Appendix D provides details about the hyperparameters used in the different experimental settings. In particular, for the experiments in Sec. 5.1, we tuned the learning rate and the model size, while we set regularization hyperparameters heuristically, as stated in Appendix D, by looking at the validation error across benchmarks. We recognize the term *model size* might be ambiguous and we will replace it with *hidden size*, which refers to the number of units in all layers in the network. For the transfer experiment (Sec. 5.3), being the same setting as in [1], we adopted the same model hyperparameters. Furthermore, we also used the same hyperparameters for the clustering and variational regularizations, while we set the hyperparameters for the other regularizations as in Sec 5.1. The experiments in Sec. 5.2 and 5.4 were carried out using model hyperparameters taken by the reference setting used in [1], while regularization hyperparameters were set to the value used in Sec. 5.1. We will improve how these details are reported, by providing an exhaustive list in the Appendix.
>
> >**Weakness 3** I am not certain about the novelty of the proposed "forgetting regularization". After a quick look into the cited papers, I fail to see significant differences to prior work and the paper does not provide a satisfying comparison.
>
> The *forget-and-relearn* paradigm refers to the general concept of resetting some of the network's weights during training. However, the choice of which weights and how/when to reset them is mostly architecture and task-specific. The novelty of our analysis concerning the *forgetting regularization* consists in applying the *forget-and-relearn* paradigm to global-local models, as a method to regularize the learning of local embeddings. The absence of comparison with the methods in [2] is due to the fact that such methods are specific to different tasks.
> As already stated, the novelty and main contribution of our work consists in being the first study and empirical analysis that systematically explores the topic of local embedding regularization and its effect in a broad number of settings. Considering previous works, [3] simply propose a clustering-based technique for local embeddings and compare it to models without local parameters in a transfer learning setting. [1] proposes a clustering regularization (similar to [3]) and a variational regularization in transfer learning, again without any comparison against other regularization methods and with the focus being the study of local components in graph-based architectures.

---

> > ### Author Response · Authors · 2024-11-21
> >
> > >**Requested change 2** The paper spends a lot of time on the definition of the models in varying generality. That space might be better used to expand the discussion of the results, the conclusion, and the discussion of the regularization techniques, in particular the proposed method.
> >
> > We think introducing the precise problem setting and the family of models is important to contextualize an empirical analysis.
> > We also kept the discussion on the results clear and direct, avoiding unsupported claims. Nonetheless, we believe results are relevant and helpful for the practitioners and provide useful guidelines as discussed when listing the findings. Note that the focus of the paper is indeed the empirical analysis of the different regularization approaches rather than proposing a new method. We will revise the paper further to make its scope clearer and improve the presentation. Moreover, we will add a paragraph with practical suggestions, summarizing what we empirically observed.
> >
> > >**Requested change 3** It is not clear, how the "simple setting" in Figure 2 relates to the benchmarks. Is there a strong reason, why global regularization should not also be considered in all evaluations as a baseline?
> >
> > The example in Figure 2 corresponds to training an STGNN on the METR-LA dataset with the hyperparameter configuration used in Sec. 5.2. We will clarify this in the figure's caption. We intend to present it as an illustrative example of how regularizing local parameters only can have a very different impact on model training than regularizing all parameters.
> > Note that we do not claim the superiority of one regularization approach over the other. Conversely, we believe they target different possible issues. In fact, in Sec. 4, we state that we expect global regularization to still be beneficial, e.g., when dealing with over-parametrized global models. As the benefits of regularizing shared weights (global regularization) are extensively recognized, we put them aside to focus on the less-studied regularization of local parameters in hybrid architectures.
> >
> > >**Requested change 4** The paper would greatly benefit from some general improvements on the writing. Here are some issues i noticed: [...]
> >
> > Thank you, we will revise the presentation and writing to account for your suggestions. Some clarifications:
> > * The encoder in Equation (3) processes each time step independently (with weights shared across all time steps and time series). We opted for a formalization that omits W to avoid confusion on this aspect.
> > * The term *embedding* is commonly used to refer to learnable parameters and positional representations in general, e.g., positional embeddings in transformers [4, 5], node embeddings in spatiotemporal graph neural networks [1], temporal embeddings for time series [6]. In our context, embeddings are vector representations associated with each individual time series analogous to the node embeddings used in spatiotemporal data. Hence, we consider the term appropriate and consistent with the existing literature.
> >
> > We thank the reviewer again for their feedback. We will upload a revised version of the paper as soon as all reviews are sent in.
> >
> > ### References
> > * [1] Andrea Cini, Ivan Marisca, Daniele Zambon, and Cesare Alippi. Taming local effects in graph-based spatiotemporal forecasting. *Advances in Neural Information Processing Systems*, 36, 2023.
> > * [2] Hattie Zhou, Ankit Vani, Hugo Larochelle, and Aaron Courville. Fortuitous forgetting in connectionist networks. *International Conference on Learning Representations*, 2021.
> > * [3] Xueyan Yin, Feifan Li, Yanming Shen, Heng Qi, and Baocai Yin. Nodetrans: A graph transfer learning approach for traffic prediction. *arXiv preprint arXiv:2207.01301*, 2022.
> > * [4] Ke Guolin, Di He, and Tie-Yan Liu. Rethinking Positional Encoding in Language Pre-training. *International Conference on Learning Representations*, 2021.
> > * [5] Heo Byeongho, Park Song, Han Dongyoon and Yun Sangdoo. Rotary position embedding for vision transformer. *European Conference on Computer Vision*, 2024.
> > * [6] Zeng Ailing, Chen Muxi, Zhang Lei and Xu Qiang. Are transformers effective for time series forecasting?. *Proceedings of the AAAI conference on artificial intelligence*, 2023.

---

### Review · Reviewer_hmzB · 2024-11-11

**Summary Of Contributions:**

This study empirically evaluates the significance of regularizing local embeddings across various regularization techniques, model architectures, and time-series forecasting datasets. In particular, regularization methods that mitigate co-adaptation between global and local parameter learning result in greater performance improvements.

**Audience:**

Yes

**Claims And Evidence:**

Yes

**Requested Changes:**

All methods and experiments are conducted within the context of time-series forecasting. Therefore, it should be emphasized more clearly in the title and Introduction section that this study focuses on forecasting, rather than on "Time Series Processing", as stated in the title.

In general, when considering the "Evaluation Criteria" of TMLR, the claims made in the submission are supported by accurate, convincing, and clear evidence. However, I have concerns regarding the limited contribution of this work, as the findings may already be widely recognized within the deep learning community. This could mean that the audience of TMLR may not find the findings of this paper particularly interesting.

**Strengths And Weaknesses:**

# Strengths
- The manuscript is well-written and easy to follow. The background preliminaries and problem settings are clearly defined, making it easier for readers to understand the objectives of this work.
- Regularization techniques for time-series forecasting are systematically studied, and the experimental analysis is thorough, including diverse model architectures and datasets.

# Weaknesses
The contributions of this work are limited, as the authors do not develop any novel regularization techniques based on their findings. Additionally, in my opinion, Findings (1) and (2) are already common knowledge and widely accepted within the deep learning community, i.e., regularization and the disentangling of global and local parameters can improve the generalization performance and robustness of deep learning systems. Furthermore, Finding (3) should not be considered a separate or novel contribution, as it essentially duplicates Finding (2).

---

> ### Author Response · Authors · 2024-11-21
>
> We thank the reviewer for their comments. In the following, we address the mentioned weaknesses and requested changes.
>
> >**Weakness 1** The contributions of this work are limited, as the authors do not develop any novel regularization techniques based on their findings. [...] Furthermore, Finding (3) should not be considered a separate or novel contribution, as it essentially duplicates Finding (2).
>
> The main objective of our paper is to show an analysis of different regularization strategies in a context where these have not been studied in depth and this is our main contribution. For what concerns Finding (3), the *forgetting* regularization shows to be among the best-performing techniques, which led us to consider Finding (3) separately. Note that resetting local parameters (and the relative network weights) to regularize the training of a related time series forecasting model has never been explored in the literature. We will improve the writing to present this contribution more clearly.
>
> >**Weakness 2** Additionally, in my opinion, Findings (1) and (2) are already common knowledge and widely accepted within the deep learning community, i.e., regularization and the disentangling of global and local parameters can improve the generalization performance and robustness of deep learning systems.
>
> While the benefits of regularization are widely accepted, our work explores different techniques for state-of-the-art forecasting models for which a systematic study on the subject is lacking. As discussed in the paper, in hybrid global-local forecasting models, local parameters are associated with each time series being forecast, and their role is very different from the one they play in other different architectures (e.g., language models). We remark that these architectures constitute the state-of-the-art for related time series forecasting and the topic is relevant in building foundation models for such task.
> Therefore, while some of the results might not be particularly surprising, they nonetheless provide practitioners with useful insights and guidelines. In our opinion, this makes our work a significant contribution.
>
> >**Requested change 1** All methods and experiments are conducted within the context of time-series forecasting. Therefore, it should be emphasized more clearly in the title and Introduction section that this study focuses on forecasting, rather than on "Time Series Processing", as stated in the title.
>
> We agree with the reviewer that the title could be more precise. We propose changing it to "On the Regularization of Learnable Embeddings for Time Series Forecasting". We will revise the Introduction accordingly. An updated version of the paper will be uploaded soon.

---

### Review · Reviewer_fGiC · 2024-11-27

**Summary Of Contributions:**

In this paper, the authors discuss how to regularize time-series specific embeddings in hybrid models which process multiple time-series at once. The authors study to which extent L1,L2-regularization, dropout, clustering-regularization, variational-regularization and random forgetting of the local embeddings help to improve performance in time-series analysis. Their main finding is, that in hybrid models, especially regularization techniques that prevent co-adaption are useful.

**Audience:**

Yes

**Claims And Evidence:**

Yes

**Requested Changes:**

- How do I read for example Table 1? Does "+ Clust" means, that I have also all regularization  above and now add "Clust", meaning I have Embeddings + L1+ L2+Clust? Or do I just have Embeddings and Clust? If it is the first, then results for having only this specific regularization are missing. If it is the latter, the results for the question "Which regularization stack up regarding performance enhancement?" are missing

- To make this work important for the community, It would be super helpful to have a section "recommendations" where it is clearly stated what kind of regularization I now should use when I am doing time-series forecasting with an own dataset

- I was not able to fully understand the co-adaption phenomenon. I get the rough idea, but it would be crucial for me to formalize it: What is a useful embedding and what is a meaningless identifier? And how can I exactly connect specific regularization methods to it?

**Strengths And Weaknesses:**

## Strengths
+ The paper is well written and easy to follow
+ Having a clear empirical evidence of what kinds of regularization are important in which scenarios is a very useful thing for the community
+ The results clearly show, that the regularizations that prevent "co-adaption" are indeed more helpful

## Weaknesses
- The term co-adaption is only explained very shallowly and the grouping of the regularization methods by the binary question wether they are preventing this is also not very formal
- What is not observed is, to which extent for adding different regularization methods the enhancements stack up, see my first request for changes for more details
- A clear recommendation how to use the findings is missing.

---

> ### Author Response · Authors · 2024-12-04
>
> We thank the reviewer for their comments. In the following, we address the mentioned weaknesses and requested changes.
>
> >**Weakness 1 and Requested change 3** The term co-adaption is only explained very shallowly and the grouping of the regularization methods by the binary question wether they are preventing this is also not very formal [...] I was not able to fully understand the co-adaption phenomenon. I get the rough idea, but it would be crucial for me to formalize it: What is a useful embedding and what is a meaningless identifier? And how can I exactly connect specific regularization methods to it?
>
> Thank you for the question, we agree that these aspects could have been better clarified. We will break down the answer into three points.
>
> >*I was not able to fully understand the co-adaption phenomenon. I get the rough idea, but it would be crucial for me to formalize it.*
>
> The issues we are discussing can be reduced to the basic (and well-formalized) concept of overfitting. We use the term co-adaptation [1] to refer to the kind of overfitting emerging from training the local embeddings end-to-end with the global model on a downstream task. If the embeddings were to be extracted separately, this kind of overfitting to the downstream model and task would not happen. In this sense, we talk about co-adaptation.
>
> >*What is a useful embedding and what is a meaningless identifier?*
>
> Again this is a contextualization of extreme overfitting (simple memorization of the training data with no utility in predicting data outside of the training set) to our target problem settings (memorization of which time series in the collection the embedding is attached to). Embeddings become more useful when they act by encoding characteristics of the dynamics of the target time series, resulting in more meaningful and (as shown in the experiments) transferable representations.
>
> >*And how can I exactly connect specific regularization methods to it?*
>
> When we talk about regularization aimed at avoiding co-adaptation, we refer to methods that prevent the downstream model from relying on specific values of the embeddings to mitigate the effect of training them end-to-end. Note that a similar design principle is behind the original dropout method: preventing co-adaptation of units in subsequent layers [1]. However, we agree with the reviewers that classifying approaches based on whether they solve an issue or not can be misleading and not correct. Therefore, in the revision, we will distinguish between approaches based on whether they perturb embeddings at training time and argue that empirical results suggest that they are more effective at preventing the kind of overfitting discussed in the above answers.
>
> We will make sure to include this discussion in the revised paper and are open to further elaborating on this if needed.
>
> >**Weakness 2 and Requested change 1** What is not observed is, to which extent for adding different regularization methods the enhancements stack up, see my first request for changes for more details [...] How do I read for example Table 1? Does "+ Clust" means, that I have also all regularization above and now add "Clust", meaning I have Embeddings + L1+ L2+Clust? Or do I just have Embeddings and Clust? If it is the first, then results for having only this specific regularization are missing. If it is the latter, the results for the question "Which regularization stack up regarding performance enhancement?" are missing
>
> Throughout the paper, we report results with each regularization applied individually. Therefore, e.g., "+ Clust" refers to applying only the *clustering* regularization to the local embeddings. We will clarify this in the table captions. We added empirical results regarding the combination of regularization pairs in Appendix F. In particular, we add L2 and *dropout* on top of *forgetting*, *clustering* and *variational* regularizations. We think this covers the most reasonable combinations, as L2 and *dropout* are commonly used in combination with other regularization approaches, while combining the others can be cumbersome. These additional results show that combining different regularizations can be beneficial; however, improvements appear mostly case-dependent, without emerging clear patterns. We thank the reviewer for the comment, as it led to additional results that can be useful for readers.

---

> > ### Author Response · Authors · 2024-12-04
> >
> > >**Weakness 3 and Requested change 2** A clear recommendation how to use the findings is missing. [...] To make this work important for the community, It would be super helpful to have a section "recommendations" where it is clearly stated what kind of regularization I now should use when I am doing time-series forecasting with an own dataset
> >
> > We agree with the reviewer that more guidelines could have been included. We have added a subsection with recommendations at the end of the experimental section.
> >
> > **References**
> > * [1] Nitish Srivastava, Geoffrey Hinton, Alex Krizhevsky, Ilya Sutskever, and Ruslan Salakhutdinov. Dropout: a simple way to prevent neural networks from overfitting. The journal of machine learning research, 15(1): 1929–1958, 2014.

---

### Author Response · Authors · 2024-12-04
**General comment on the revision**

We thank all reviewers again for their comments which led to improvements to our work. These can be found in the revised version of the document we just uploaded. Relevant changes made to the text are highlighted in purple.

We summarize the changes as follows:
* we added experimental results on the combination of different regularization strategies in Appendix F;
* we added Section 5.5, for the purpose of offering clear and concise guidelines (practical suggestions) for readers;
* we clarified the meaning of *co-adaptation* and refined the distinction between regularization approaches;
* we clarified that resetting local parameters was not previously studied in this context and clarified its role in supporting Finding (3);
* we improved Section 5 and re-wrote Appendix D to provide more precise details on the experimental settings and on the models' hyperparameters;
* we changed the title to "On the Regularization of Learnable Embeddings for Time Series Forecasting" to reflect more appropriately the scope of our study;
* we rephrased our findings to align more appropriately with the observed results;
* we brought improvements to the notation as highlighted by the reviewers and made some additional considerations on the empirical results, as well as several minor modifications to the writing in general.

We thank again the AE and the reviewers for their time. To the reviewers, please let us know if we addressed your concerns. We remain open for further discussion.

---

### Decision · Action_Editor_nYSh · 2025-01-27

**Recommendation:** Accept as is

**Comment:**

All reviewers expressed the opinion that the contributions of the work are limited, and two of the reviewers provided a recommendation of “leaning to reject”.

Having said this, two of the reviewers did acknowledge that the submission did meet the acceptance criteria of TMLR, namely that the claims of the submission are supported by clear evidence, and the results will be of interest to some individuals in TMLR’s audience.

The third reviewer is of the opinion that claims are not sufficiently supported because the paper lacks a comprehensive comparison to global regularization methods and the number of evaluated models is too small. Unfortunately, this reviewer did not express in the original review that the number of evaluated models is too small. While the results would be more convincing with more models, the experiments do include state-of-the-art methods.

Concerning the comparison to global regularization methods, the authors provided a response to this, explaining that the benefits of global regularization are well known and for this reason the paper focuses on the less-studied issue of regularization of the local parameters. The reviewer did not respond to or acknowledge the authors’ response to the criticism. The official recommendation does not provide a clear argument about why this response is unsatisfactory.

In my view, the paper’s experiments do support the claims that are put forward in the paper, provided they are read in the context of “for the models that we experiment with”, which is implied in the introduction.

Concerning audience interest, the reviewers expressed concerns that the contribution is limited, and that the results may be either common knowledge or would emerge through standard hyperparameter tuning. In general, however, there is a consensus that the results for such an investigation have not been published for local parameters of time-series models, and that some researchers working on time-series analysis and forecasting are likely to be interested in the findings.

**Audience:**

Some individuals in TMLR's audience are likely to be interested in knowing the findings of the paper.

**Claims And Evidence:**

The claims made in the submission are supported by sufficient evidence, which is accurate, convincing and clear.